

# The sensitivity of Southern Ocean atmospheric dimethyl sulfide to modelled sources and emissions

Yusuf A. Bhatti[1], Laura E. Revell[1], Alex J. Schuddeboom[1], Adrian J. McDonald[1,2], Alex T. Archibald[3,4], Jonny Williams[5], Abhijith U. Venugopal[1], Catherine Hardacre[6], and Erik Behrens[5]

[1]School of Physical and Chemical Sciences, University of Canterbury, Christchurch, New Zealand
[2]Gateway Antarctica, University of Canterbury, Christchurch, New Zealand
[3]National Centre for Atmospheric Science, Cambridge, United Kingdom
[4]Yusuf Hamied Department of Chemistry, University of Cambridge, Cambridge, United Kingdom
[5]National Institute of Water and Atmospheric Research (NIWA), Wellington, New Zealand
[6]Met Office, Exeter, EX1 3PB, United Kingdom

**Correspondence:** Yusuf Bhatti (yusuf.bhatti@pg.canterbury.ac.nz)

**Abstract.** The biogeochemical behaviour of the Southern Ocean is complex and dynamic. The processes that affect this behaviour are highly dependent on physical, chemical, and biological constraints, which are poorly constrained in Earth System Models. We assess how emissions of dimethyl sulfide (DMS), a precursor of sulfate aerosol, change over the Southern Ocean when the chlorophyll-a distribution, which influences oceanic DMS production, is altered. Using a nudged configuration of the atmosphere-only United Kingdom Earth System Model, UKESM1-AMIP, we performed nine 10-year simulations using forcings representative of the period 2009 – 2018. Four different seawater DMS data sets are tested as input for these simulations. Three different DMS sea-to-air flux parameterizations are also explored. Our goal is to evaluate the changes in oceanic DMS, sea-to-air fluxes of DMS, and atmospheric DMS through these different simulations during austral summer. The mean spread across all the simulations with different oceanic DMS datasets, but the same sea-to-air flux parameterizations, is 112% (3.3 to 6.9 TgS Yr$^{-1}$). The mean spread in simulations using the same oceanic DMS dataset, but differing sea-to-air flux parameterisations is 50-60% (2.9 to 4.7 TgS Yr$^{-1}$). The choice of DMS emission parameterisation has a larger influence on atmospheric DMS than the choice of oceanic DMS source. We also find that linear relationships between wind and DMS flux generally compare better to observations than quadratic relationships. Simulations that implement a quadratic emission rate show on average 35% higher DMS mixing ratios than the linear emission rates. Simulations using seawater DMS derived from satellite chlorophyll-a data in combination with a recently-developed flux parameterisation for DMS show the closest agreement with atmospheric DMS observations and are recommended to be included in future simulations. This work recommends for Earth System Models to include a sea-to-air parameterization that is appropriate for DMS, and for oceanic DMS datasets to include inter-annual variability based on observed marine biogenic activity. Such improvements will provide a more accurate process-based representation of oceanic and atmospheric DMS, and therefore sulfate aerosol, in the Southern Ocean region.



## 1 Introduction

The representation of aerosols over the Southern Ocean is a large source of uncertainty in climate models due to the lack of observational data and large seasonal variability (Revell et al., 2019). Poor representation of aerosols contributes to the large biases in future climate projections over the Southern Ocean (Myhre et al., 2014). Sea spray and dimethyl sulfide (DMS; $CH_3SCH_3$) are fundamental sources for aerosol formation over this region (Revell et al., 2021; Bhatti et al., 2022). The domi-

nant source of sulfate over the marine atmosphere is the biogenic marine aerosol precursor DMS, controlled by phytoplankton productivity (Keller et al., 1989; Bates et al., 1987; Berndt et al., 2019). Revell et al. (2019) found sulfate aerosol production from DMS was responsible for around 60% of the austral summer aerosol optical depth over the Southern Ocean. Atmospheric DMS therefore has the potential to greatly influence cloud condensation nuclei during austral summer, due to its high rate of emissions (Kloster et al., 2006; Revell et al., 2019; Korhonen et al., 2008; Pandis et al., 1994).

The Southern Ocean contains extremely high phytoplankton productivity during austral summer (December, January, and February) (Deppeler and Davidson, 2017). Phytoplankton activity plays a key role in chlorophyll-$a$ (chl-$a$) production and is considered to be a key driver of oceanic DMS production (e.g. Uhlig et al., 2019; Townsend and Keller, 1996; Anderson et al., 2001; Deppeler and Davidson, 2017). Earth System Models represent the process of oceanic DMS formation through multiple mechanisms, with varying focus on chl-$a$, nutrients, light, mixed-layer depth, zooplankton, and dimethylsulfoniopropionate

(Bock et al., 2021). The UKESM1 and MIROC-ES2L use a diagnostic approach to represent chl-$a$ (Sellar et al., 2019; Anderson et al., 2001; Hajima et al., 2020). CNRM-ESM2-1 and NorESM2-LM focus on a prognostic approach, closely related to zooplankton and dimethylsulfoniopropionate, precursors of oceanic DMS (Seland et al., 2020; Séférian et al., 2019). CMIP6 models simulate biases in oceanic DMS production compared with observational climatologies of DMS in the Southern Ocean region (Bock et al., 2021).

Atmosphere-only climate models use climatologies to approximate the global concentration of oceanic DMS. Lana et al. (2011) and Kettle et al. (1999) constructed observational climatologies of oceanic DMS which are used within climate models. However, there is a limited amount of data available within the Southern Ocean, which can lead to biases when compared to other regions (e.g. Bock et al., 2021; Mulcahy et al., 2020). A limitation of representing oceanic DMS as a static climatology is that it does not account for the large temporal variations in DMS concentrations observed. For instance, El Niño Southern

Oscillation (ENSO) events, wildfires, and volcanic eruptions all significantly influence oceanic DMS within the Southern Ocean (e.g. Yoder and Kennelly, 2003; Tang et al., 2021; Wang et al., 2022; Browning et al., 2015; Longman et al., 2022). Calculating oceanic DMS online using a biological proxy would resolve these perturbing events (Galí et al., 2018).

DMS is emitted from the ocean to the atmosphere and has a strong dependence on the surface wind speed (e.g. Fairall et al., 2011). A wealth of research has focused on better understanding the relationship between atmospheric DMS and wind speed

(Vlahos and Monahan, 2009; Zavarsky et al., 2018; Blomquist et al., 2017; Wanninkhof, 1992, 2014; Nightingale et al., 2000; Liss and Merlivat, 1986; Goddijn-Murphy et al., 2016; Ho et al., 2006; Bell et al., 2015). However, the uncertainty in this relationship remains high particularly within the Southern Ocean due to a lack of observational data (e.g. Elliott, 2009; Smith et al., 2018; Zhang et al., 2020), particularly for wind speeds $\geq 13$ ms$^{-1}$ (Blomquist et al., 2017). Recently, significant progress




has been made as recent literature has established that DMS flux has a linear relationship with wind (Goddijn-Murphy et al.,
2016; Blomquist et al., 2017; Bell et al., 2015), while Earth System Models continue to use older quadratic relationships to
represent DMS emissions Bock et al. (2021).

   Oceanic DMS observations in the Southern Ocean are highly variable in time and space (Lana et al., 2011; Hulswar et al.,
2022; Galí et al., 2018), while the emissions of DMS are also uncertain (e.g. Korhonen et al., 2008; Blomquist et al., 2017).
This study sets out to examine whether including oceanic DMS with spatio-temporal variability based on real-world chl-$a$
observations improves the simulation of atmospheric DMS. We investigate differences in oceanic DMS and emission param-
eterizations for forming atmospheric DMS using the nudged to observation configuration of UKESM1-AMIP. We calculate
a 10-year monthly time series calculated from chl-$a$ MODIS-aqua satellite data implemented within the modified Anderson
et al. (2001) parameterization (Sellar et al., 2019). We also test climatologies from Lana et al. (2011), Hulswar et al. (2022),
and the DMS climatology used by UKESM1-AMIP (Sellar et al., 2019). DMS emissions are calculated using two quadratic
(Wanninkhof, 2014; Nightingale et al., 2000) and two linear (Liss and Merlivat, 1986; Blomquist et al., 2017) sea-to-air flux
parameterizations. Evaluating the process and sensitivity of DMS from the ocean to the atmosphere in climate models is crit-
ical for the further development of models and for understanding the biogeochemical cycle. We compare the DMS variability
across the Southern Ocean during summer, improving our understanding of the relative importance of choosing the source
(oceanic DMS) and emissions.

## 2   Methods

### 2.1   Model Configuration

Simulations were performed using the atmosphere-only configuration of the coupled UK Earth System Model (UKESM1;
Yool et al., 2020; Sellar et al., 2019; Mulcahy et al., 2020). UKESM1 simulates ocean biogeochemistry via an intermediate
complexity biogeochemical dynamic model, MEDUSA2.0 (the Model of Ecosystem Dynamics, nutrient Utilization, Seques-
tration, and Acidification;  Yool et al., 2020, 2013). MEDUSA is used in UKESM1 to represent biogeochemical feedbacks
within the Nucleus for European Modelling of the Ocean (NEMO) ocean model (Madec et al., 2008). The aerosol component
of UKESM1 uses the GLOMAP-mode aerosol scheme, which is described in full by Mulcahy et al. (2020) and Mann et al.
(2010, 2012). Wind and temperatures within the simulations used in this study are nudged 6-hourly to real-world conditions
via the use of the ERA-5 reanalysis data (Hersbach et al., 2020). The full description of how nudging is incorporated within the
UKESM1-AMIP is outlined in more detail by Telford et al. (2008). As noted by Pithan et al. (2022) and Kuma et al. (2020),
nudging simulations can enhance the precision of simulations used for assessing atmospheric processes. Specifically, it allows
for a more accurate representation of meteorological factors such as wind speed, which play a key role in the formation of
atmospheric DMS. Using nudged runs also allows us to better evaluate our simulations against observational measurements
made during voyages.





**Table 1.** Oceanic DMS data sets used in the model simulations.

| Oceanic DMS dataset | Source | Citation | Year of Data |
|---|---|---|---|
| Lana | Oceanic DMS observations | Lana et al. (2011) | 1972 - 2009 |
| Hulswar | Oceanic DMS observations | Hulswar et al. (2022) | 1972 - 2021 |
| MEDUSA | UKESM1 CMIP6 simulations | Anderson et al. (2001); Sellar et al. (2019) | 1979 - 2014 |
| MODIS-DMS | MODIS-aqua chlorophyll-$a$ via Anderson et al. (2001) | N/A (produced for this study) | 2009 - 2018 |

The sea-to-air transfer of DMS in our simulations is discussed in Section 2.3. All simulations in this study are 10 years long, spanning 2009 to 2018. We focus on the austral summer months (December–February; DJF) due to the summer being the most biologically productive season.

In this paper, we compare observational data to our simulations using the same hourly timescales. To evaluate variability, we use the coefficient of variation (CoV) which is a statistical measure that compares the variability of data by expressing the

standard deviation as a percentage of the mean. CoV is used to compare the variability between each of the simulations oceanic DMS, DMS emissions, and atmospheric DMS concentration. A higher CoV suggests that the variability or dispersion of the data is relatively large compared to its mean. Where uncertainty is reported, 1 standard deviation through time and space is stated.

## 2.2 Oceanic DMS

We input four oceanic DMS data sets into the atmospheric model: three climatologies and one 10-year time-series between 2009 to 2018. Two are observational-based climatologies from Lana et al. (2011) (hereafter 'Lana') and Hulswar et al. (2022) (hereafter 'Hulswar'). The 'MEDUSA' climatology (1979-2014) is sourced from the UKESM1 CMIP6 repository, MEDUSA (Yool et al., 2021; Sellar et al., 2019; Tang et al., 2019). See Table 1 for an outline of the oceanic DMS climatologies and dataset used.

The UKESM1 uses a diagnostic approach in the formulation of oceanic DMS, which is calculated online using surface daily shortwave radiation ($J$), dissolved inorganic nitrogen ($Q$), and surface chl-$a$ ($C$):

$$Oceanic\ DMS = a,\ for\ log(CJQ) \leq s \tag{1}$$

$$Oceanic\ DMS = b[log(CJQ) - s] + 1,\ for\ log(CJQ) > s \tag{2}$$

The fitted parameter values are a=1, b=8, and s=1.56, as described by Sellar et al. (2019). The online oceanic DMS from MEDUSA in the UKESM1 shows small annual variability and therefore a 30-year climatology will represent MEDUSA well.



$Q$ and chl-$a$ are taken from MEDUSA, and $J$ is from the atmosphere component of the UKESM1, the Unified Model. Chl-$a$ is used to calculate oceanic DMS concentrations in other CMIP6 models, such as MIROC-ES2L, and within algorithms such as that detailed by Galí et al. (2018). The Anderson et al. (2001) parameterization is a widely used and well-validated method for calculating oceanic DMS in UKESM1. Here, we have tested a modified version of it using the MODIS-aqua chl-$a$ dataset. This data set, 'MODIS-DMS', is a continuous time series between 2009 to 2018. MODIS-DMS is calculated offline using the same diagnostic parameterization (Anderson et al., 2001; Sellar et al., 2019) as Equations 1 and 2. The UKESM1 has a +6 W m$^{-2}$ bias for $J$ within CMIP6 over the Southern Ocean, which may result in slightly higher oceanic DMS concentrations (Schuddeboom and McDonald, 2021). The $J$ and $Q$ used to calculate MODIS-DMS remain the same to MEDUSA, but a new monthly-mean chl-$a$ field ($C$) is introduced via Moderate Resolution Imaging Spectroradiometer (MODIS) -Aqua Level-3 ocean-color chl-$a$ (Table 1; e.g. Hu et al., 2019; O'Reilly and Werdell, 2019). Bi-linear interpolation is used to fill in small gaps (around 1% for monthly averages) of spatial chl-$a$ data. Using the MODIS-Aqua chl-$a$ satellite data, oceanic DMS concentrations were calculated each month for our 10 year period. From this, we capture the annual variability of the distribution in ocean biological productivity (referenced in this work as MODIS-DMS). Our goal is to understand the relationship between oceanic biological productivity, as represented by chl-$a$, and atmospheric DMS concentrations in the Southern Ocean during austral summer. We then evaluate which oceanic DMS sources produce the best distribution compared to observations.

Several studies have validated the MODIS-aqua Ocean Color chl-$a$ retrieval, finding it to generally underestimate Southern Ocean conditions (Zeng et al., 2016; Haëntjens et al., 2017; Jena, 2017). Satellites can also overestimate chl-$a$ measurements due to the scattering of light from aerosols (Schollaert et al., 2003). However, Marrari et al. (2006) found satellite chl-$a$ is accurate within the Southern Ocean during summer. Therefore the high spatial and temporal availability of summertime data makes chl-$a$ a viable option for estimating phytoplankton productivity and oceanic DMS. Using the MODIS-DMS data set, we aim to accurately simulate maxima and minima in oceanic DMS concentrations comparable to observations. By comparing the model results with observations, we can identify and understand the impact of phytoplankton bloom events and annual variability which can not be captured by climatologies.

## 2.3 DMS Sea-to-Air Flux

To calculate the transfer of DMS from the ocean to the atmosphere, a parameterization is used which is controlled by wind speed. The formulation of the transfer velocity is derived from observational measurements of a particular gas. Many flux parameterisations have been developed, but these have mostly been based on gases such as $CO_2$ (e.g. Wanninkhof, 2014). These parameterizations are widely implemented within climate models to represent DMS but vary depending on the model. We tested three flux parameterisations shown in Figure 1. Blomquist et al. (2017) (hereafter 'B17') used DMS measurements to derive a relationship between wind speed and DMS, whereas Wanninkhof (2014) (W14) and Liss and Merlivat (1986) (LM86) used $CO_2$, and other high solubility gases. Sea-to-air parameterizations are typically linear or quadratic, depending on the solubility of the gas. Linear equations best represent gases with intermediate solubilities, such as DMS (e.g. Blomquist et al., 2017; Goddijn-Murphy et al., 2016; Bell et al., 2015), while quadratic equations are better suited for highly soluble gases like



**Table 2.** Simulations used in this study, with the oceanic DMS data sets as the name, followed by the DMS flux parameterization used.

| Simulation name | Oceanic DMS source | DMS flux parameterization |
|---|---|---|
| Lana$_{LM86}$ | Lana et al. (2011) | Liss and Merlivat (1986) |
| Lana$_{B17}$ | Lana et al. (2011) | Blomquist et al. (2017) |
| Lana$_{W14}$ | Lana et al. (2011) | Wanninkhof (2014) |
| Hulswar$_{LM86}$ | Hulswar et al. (2022) | Liss and Merlivat (1986) |
| MEDUSA$_{LM86}$ | Anderson et al. (2001); Sellar et al. (2019) | Liss and Merlivat (1986) |
| MODIS$_{LM86}$ | N/A (produced for this study) | Liss and Merlivat (1986) |
| MODIS$_{B17}$ | N/A (produced for this study) | Blomquist et al. (2017) |
| MODIS$_{W14}$ | N/A (produced for this study) | Wanninkhof (2014) |

$CO_2$ (Wanninkhof, 2014; Nightingale et al., 2000; Wanninkhof, 1992). This study uses two linear equations from LM86 and B17 to represent DMS emissions more accurately compared with observations, as suggested by Blomquist et al. (2017) and Goddijn-Murphy et al. (2016). LM86 is a piecewise function consisting of three lines with different gradients and intercepts, depending on the wind speed (Figure 1). LM86 is used as the default flux parameterization within the UKESM1 (Sellar et al., 2019) and is thus used on with oceanic DMS datasets. The quadratic formula from Wanninkhof (2014) is also tested. Using these different parameterizations provides an estimate of the spread of DMS emissions. The Lana oceanic DMS climatology is tested with the W14, and B17 fluxes, as Lana is currently the most widely used climatology within climate models (Bhatti et al., 2022). We also apply the W14 and B17 flux parameterisations to the MODIS-DMS oceanic DMS dataset to test the lower limits of oceanic DMS concentrations to assess the variation from the time-series. Table 2 outlines the sensitivity simulations performed for this study, described by the oceanic DMS concentration subscripted with the sea-to-air flux used. For example, Lana$_{LM86}$ means that the simulation used the Lana et al. (2011) climatology as its oceanic DMS source, and the DMS flux parameterisation of Liss and Merlivat (1986).

To calculate the flux of DMS, the Schmidt number of DMS is required. The Schmidt number describes the mixing efficiency of a substance in a fluid and is used to calculate the transfer velocity of gas from the sea to air. We update the Schmidt number of DMS ($Sc_{DMS}$) used in the UKESM1 from the formulation used in Saltzman et al. (1993) to Wanninkhof (2014), as shown in Equation 3:

$$Sc_{DMS} = 2855.7 + (-177.63 + (6.0438 + (-0.11645 + 0.00094743 \cdot T) \cdot T) \cdot T) \cdot T \qquad (3)$$

T is the sea surface temperature derived from The Hadley Centre Global Sea Ice and Sea Surface Temperature (HadISST) within the model (Titchner and Rayner, 2014). LM86 was constructed based on gases other than DMS, but is often used for DMS emissions within CMIP6 Earth System and climate models (e.g. Krasting et al., 2018; Tang et al., 2019; Ridley et al., 2019; Yukimoto et al., 2019). In equation 4, $U_{10}$ is the wind speed at 10 m above the surface and $K_w$ represents the transfer velocity of DMS:



for $u_{10} \leq 3.6$ :

$$K_w = 0.17 \left( \frac{600}{Sc_{DMS}} \right)^{\frac{2}{3}} u_{10},$$

for $3.6 \leq u_{10} < 13$ :

$$K_w = 2.85 \left( \frac{600}{Sc_{DMS}} \right)^{\frac{1}{2}} (u_{10} - 3.6) + 0.612 \left( \frac{600}{Sc_{DMS}} \right)^{\frac{2}{3}},$$

for $u_{10} > 13$ :

$$K_w = 5.9(u_{10} - 13) \left( \frac{600}{Sc_{DMS}} \right)^{\frac{1}{2}} + 26.79(u_{10} - 3.6) \left( \frac{600}{Sc_{DMS}} \right)^{\frac{1}{2}} + 0.612 \left( \frac{600}{Sc_{DMS}} \right)^{\frac{2}{3}} \tag{4}$$

W14 uses a quadratic formula (equation 5) to empirically fit observations of $CO_2$ as a sea-to-air transfer. W14 is also very frequently used to calculate DMS emissions amongst CMIP6 simulations (e.g. Tjiputra et al., 2020).

$$K_w = 0.251 \cdot u_{10}^2 \left( \frac{660}{Sc_{DMS}} \right)^{\frac{1}{2}} \tag{5}$$

Finally, B17 is the only parameterization used in this study which calculates a transfer velocity based on real-world observation of DMS (equation 6). B17 is a superlinear and sub-quadratic parameterization, however, for simplicity and the wind speeds used in this study, we label B17 as a linear parameterization.

$$K_w = 0.7432 \cdot u_{10}^{1.33} \left( \frac{660}{Sc_{DMS}} \right)^{\frac{1}{2}} \tag{6}$$

### 2.4 Observational Datasets

Two Southern Ocean voyages are used to validate our simulations: the SOAP (Surface Ocean Aerosol Production; Bell et al., 2015; Law et al., 2017) campaign and RV Tangaroa voyage (TAN1802; Kremser et al., 2021). The SOAP voyage measured oceanic and atmospheric DMS from Feb-March 2012 near the Chatham Rise (within 42–47 °S, 172–180 °E) off the east coast of New Zealand, a highly biologically productive region of the Southern Ocean (Bell et al., 2015; Smith et al., 2018). The TAN1802 voyage measured oceanic DMS along a transect in the Southern Ocean during Feb-March 2018 between latitudes 40 °S to 70 °S, 180 °E (Kremser et al., 2021). Other voyages outside the years covered by our nudged simulations, but included in the atmospheric DMS analysis are the SOIREE and ANDREXII voyages, used to calculate the observational atmospheric DMS. SOIREE occurred in Feb 1999 and measured atmospheric DMS concentration (Boyd and Law, 2001) between 42 - 63 °S, 139–172 °E. ANDREXII (Wohl et al., 2020) travelled longitudinally around 60 °S, between February to April 2019.

We used oceanic DMS measurements for TAN1802 Kremser et al. (2021), SOAP (Bell et al., 2015), and ERA-5 surface wind speeds (Hersbach et al., 2020) to calculate hourly DMS emissions. The Wanninkhof (2014) DMS Schmidt number is



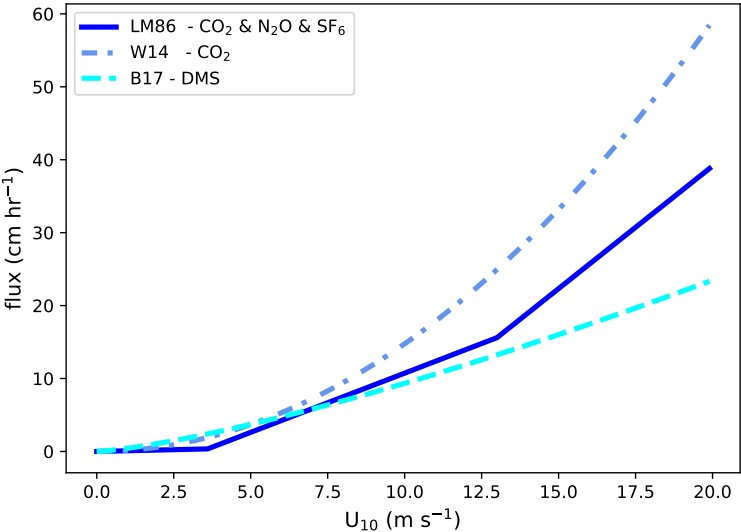

**Figure 1.** DMS sea-to-air flux parameterizations tested in this study. LM86 = Liss and Merlivat (1986); W14 = Wanninkhof (2014) and B17 represents Blomquist et al. (2017). The gases labelled in the legend are the measurements taken to identify the gas exchange relationship.

calculated using the same parameters used within the simulations, for consistency with comparisons to simulated fluxes. Sea ice and sea surface temperature data are from the Met Office Hadley Centre's sea ice and sea surface temperature (HadISST; Titchner and Rayner, 2014), where sea surface temperature represents T in Equation 3. The HadISST and ERA-5 wind speed data were obtained for the same time and location as the two voyages (within the nearest neighbour grid cell). We applied three different sea-to-air flux parameterizations (LM86, B17, and W14) to both SOAP and TAN1802 voyage paths (See section 3.2).

We compare our simulations to the voyage dataset using the hourly model output and identify the nearest neighbour grid cell to the ship location. Analysis of oceanic DMS data used in the models is also synchronized to TAN1802 and SOAP voyages, using the same timescales for comparing the voyages with model data.

We also validate the model using atmospheric DMS concentrations measured at three stations: Cape Grim (1989 to 1996; 41 °S and 145 °E); Amsterdam Island (1987 to 2008; 38 °S, 78 °E); and King Sejong Station (2018 to 2020; 62 °S, 58 °W). King Sejong is located on the Antarctic Peninsula, where sea ice melt occurs during our study period, which can profoundly increase DMS emissions, as previously found by Berresheim et al. (1998); Read et al. (2008). The climatologies from Amsterdam Island and Cape Grim stations are compared with the model climatology of atmospheric DMS. The King Sejong measurements align with our simulation period, and so we compare both datasets on the same timescale.





## 3 Results

### 3.1 Oceanic DMS

Figure 2a-d shows the spatial distribution of the oceanic DMS from the different datasets used in this study. Each distribution
has key defining characteristics, although Hulswar (Figure 2 d) is similar to Lana (Figure 2 c) as it is an updated version.
When the dataset includes chlorophyll-$a$ (chl-$a$), oceanic DMS has distinguishable features across latitudes, partly due to the
influence of the Southern Hemisphere westerly jet, driving ocean circulation and transporting phytoplankton (e.g. Allison et al.,
2010; Li et al., 2016). The only difference between the calculation of MODIS-DMS and MEDUSA oceanic DMS is the chl-$a$
input, however, their distributions of oceanic DMS in the Southern Ocean are largely different, as illustrated in Figure 2e.
Observational-based climatologies, such as in Lana or Hulswar (Figure 2c,d), do not consider other proxies of oceanic DMS
(Lana et al., 2011; Hulswar et al., 2022). Lana and Hulswar (Figure 2c,d), do not match the distribution of chl-$a$ in the Southern
Ocean, particularly along the Antarctic Circumpolar Current, as oceanic DMS concentrations are focused to a specific region,
based only on observations of oceanic DMS (Lana et al., 2011; Hulswar et al., 2022). The difference between the mean of
MODIS and MEDUSA (the lowest and highest mean of all the oceanic DMS datasets used) is 107%, respectively.

MEDUSA produces the most homogeneous oceanic DMS distribution in the summertime Southern Ocean, with the highest
mean of 4.88 nM. Additionally, it has the smallest standard deviation of $\pm0.87$ nM (and the lowest CoV of $\pm17\%$ indicating a
small spread of variance). The chl-$a$ calculated by MEDUSA has a positive bias when compared to observations in the Southern
Ocean during summer (Yool et al., 2013, 2021), resulting in higher oceanic DMS concentrations than other datasets. In contrast,
the MODIS-DMS dataset produces low oceanic DMS concentrations in open ocean regions, but very high concentrations in
biologically productive regions (near the subtropical front), such as the Chatham Rise and coastal South America (Behrens and
Bostock, 2023). MODIS-DMS exhibits large variability due to locally-enhanced chl-$a$ concentrations along coastal regions
and the mid-latitudes (40-50 °S) of the Southern Ocean. Oceanic DMS from MODIS has a mean of 2.36$\pm$1.57 nM (CoV of
67%), which is outside the range of oceanic DMS produced by MEDUSA, highlighting the sensitivity of the Anderson et al.
(2001) parameterization to the chl-$a$ concentration.

MODIS-DMS oceanic DMS concentrations vary each summertime across the Southern Ocean during the 10 year climatol-
ogy (See Figure A1 in the supplementary materials). The year with the highest mean oceanic DMS concentration observed
by the MODIS-DMS dataset (2.58$\pm$2.12 nM) occurred in 2010 (Figure A1), with a 16.2% higher concentration than the low-
est concentration in 2015 (2.22$\pm$1.88 nM). The largest interannual variability in MODIS-DMS occurs around New Zealand
and the East Coast of South America and is likely caused by specific phytoplankton bloom events, possibly being influenced
by ENSO (e.g. Santoso et al., 2017; Thompson et al., 2015; Yoder and Kennelly, 2003). Oceanic DMS climatologies do not
capture these inter-annual oceanic events. Furthermore, voyages that measure oceanic DMS often have specific research tar-
gets which can cause a sampling bias within the climatologies compiled from *in-situ* observations. Voyages also only collect
data during specific months within specific regions. For example, the SOAP voyage targeted phytoplankton blooms and their
accompanying high oceanic DMS concentrations (Bell et al., 2015).



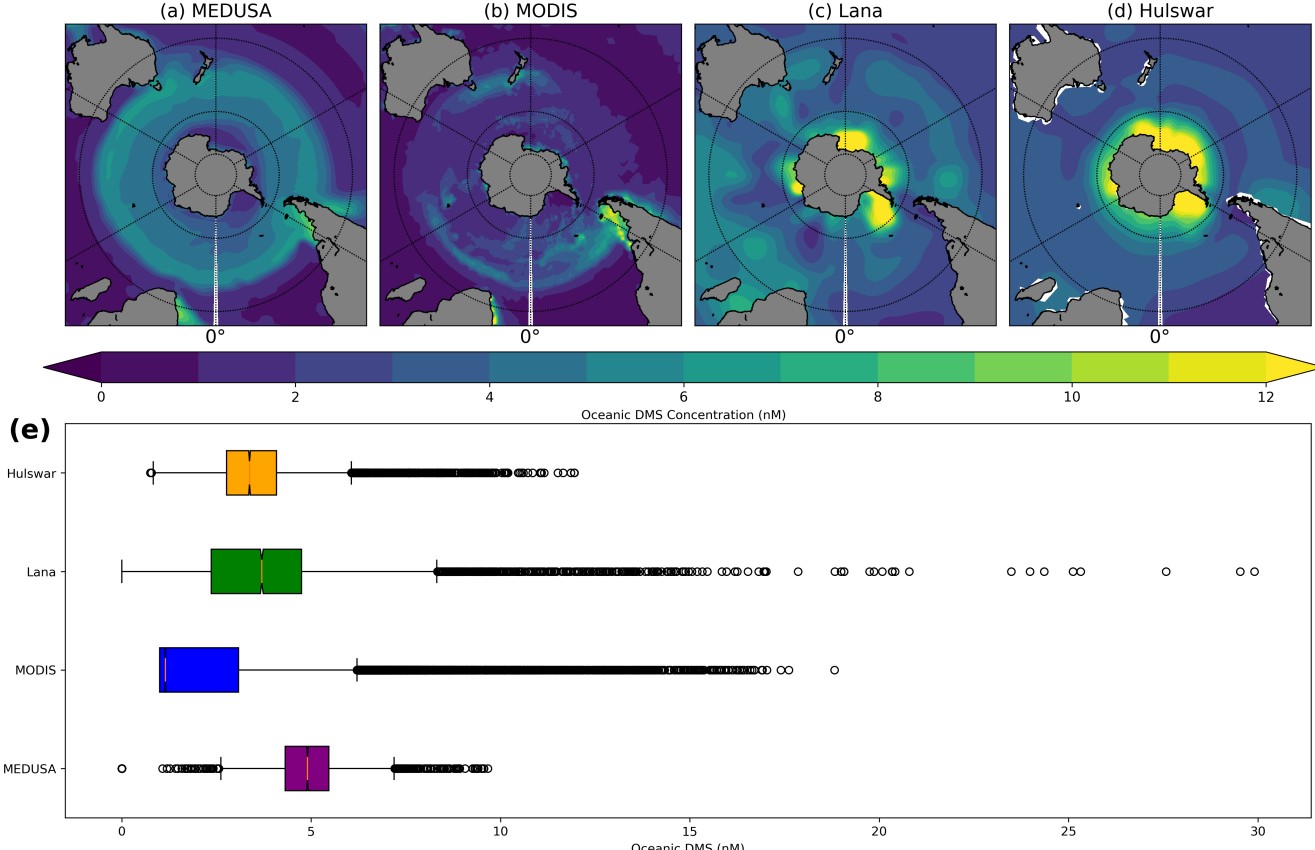

**Figure 2.** summertime (DJF) Oceanic DMS in the Southern Ocean (40 - 60 °S). The spatial distribution (a-d) shows the (a) UKESM1 climatology from MEDUSA, (b) the climatology from MODIS-DMS, and observational-based climatologies of (c) Lana and (d) Hulswar. (e) The box plot shows the distribution of each oceanic DMS dataset used. The data points outside the whiskers represent 0.7% of the dataset, highlighting the outliers of the distribution.

230  Factors such as melting sea ice can also affect chl-$a$, and therefore oceanic DMS (Behera et al., 2020; Berresheim et al., 1998). Phytoplankton activity, such as bloom events, affect chl-$a$ concentrations (e.g. Uhlig et al., 2019; Matrai et al., 1993) and will be captured by the MODIS-DMS simulations, but not by the climatologies; MEDUSA currently lacks the ability to represent realistic phytoplankton blooms in chl-$a$ concentrations (Yool et al., 2021).

Lana and Hulswar have similar means and CoV, respectively, across the entire Southern Ocean during austral summer
235  (3.87 nM and 3.51 nM; CoV of 31% and 32%). However, the distribution of both datasets (Figure 2e) is different: Lana contains much higher concentrations, maximizing at 30 nM compared to 14 nM from Hulswar. Using the Anderson et al. (2001) parameterization while changing the chl-$a$ input, MEDUSA calculates a peak DMS concentration at 11 nM, whereas MODIS-DMS is 64% greater, maximizing at 18 nM.





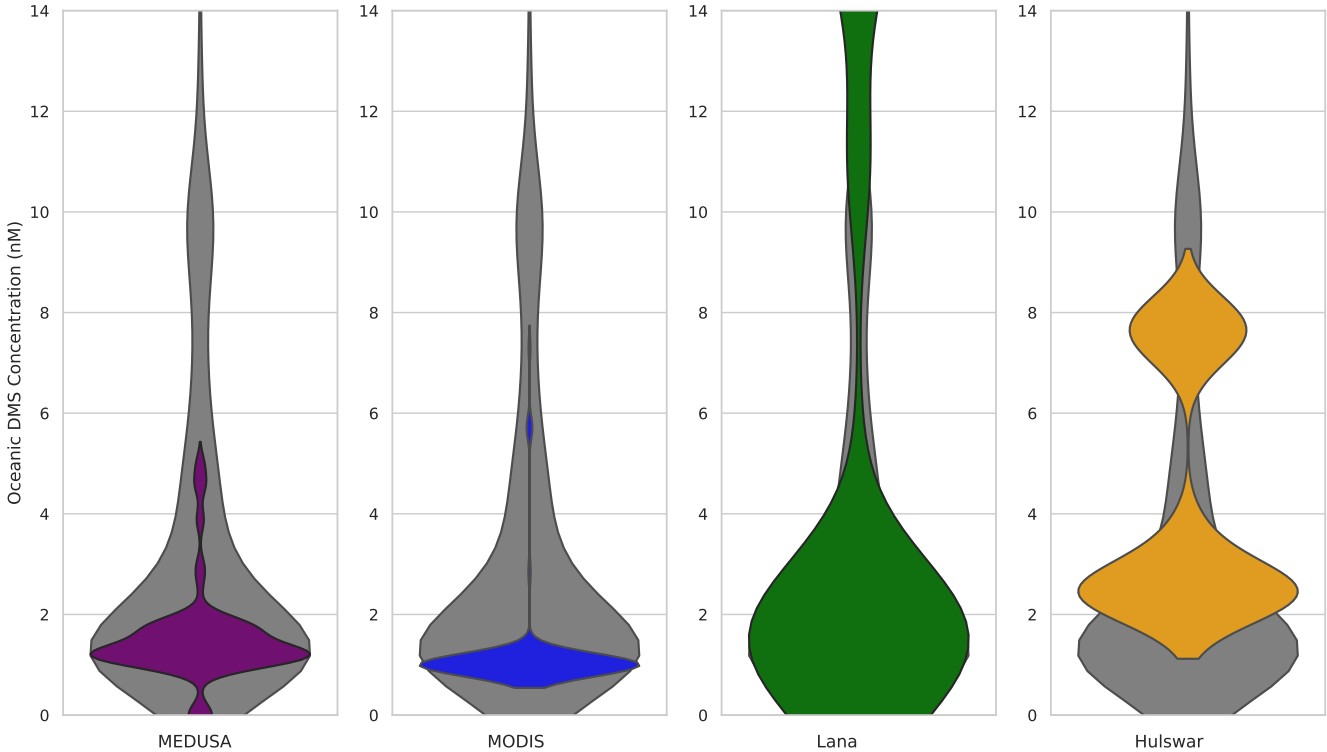

**Figure 3.** Violin plots of TAN1802 data. Overlaid are the oceanic DMS datasets used in the model simulations (Feb to March 2018, 40 °S to 70 °S, 180 °E).

By examining localized oceanic DMS measurements within the Southern Ocean obtained during the TAN1802 (Figure 3) and SOAP (Figure 4) voyages in comparison to each model input data, we can determine the variations across each simulation. The oceanic DMS from the model overlays the respective voyage data (grey) in Figure 3 and Figure 4. Lana fits the distribution of TAN1802 more closely than the other datasets, as illustrated by the higher DMS concentrations. The differences between the two climatologies are a result of additional observational datasets within Hulswar. MODIS-DMS and MEDUSA have the lowest means, 1.19 and 1.52 nM respectively, but MODIS-DMS has a higher CoV of 79% due to higher concentrations at lower latitudes (45 °S) of the Southern Ocean. TAN1802 has a CoV of 105%, similar to Lana's 114%. Hulswar overestimates DMS concentrations by a factor of two between 45 and 65 °S. Observation-based climatologies capture high oceanic DMS concentrations better than parameterization-based concentrations, as illustrated by the violin plot in Figure 3.

SOAP voyage data represents oceanic DMS concentrations during phytoplankton bloom events, therefore the shape of the observed DMS distribution (Fig. 4) is quite different to the TAN1802 data (Fig. 3) and would be expected to be highly biased. All of the oceanic DMS datasets fail to capture the higher concentrations measured by SOAP, displaying a positively skewed distribution (Bell et al., 2015); with most concentrations clustered between 2-4 nM. MODIS-DMS has the greatest variability (CoV of 36%), highest average, and largest maximum concentration. MODIS-DMS also has the best linear relationship with





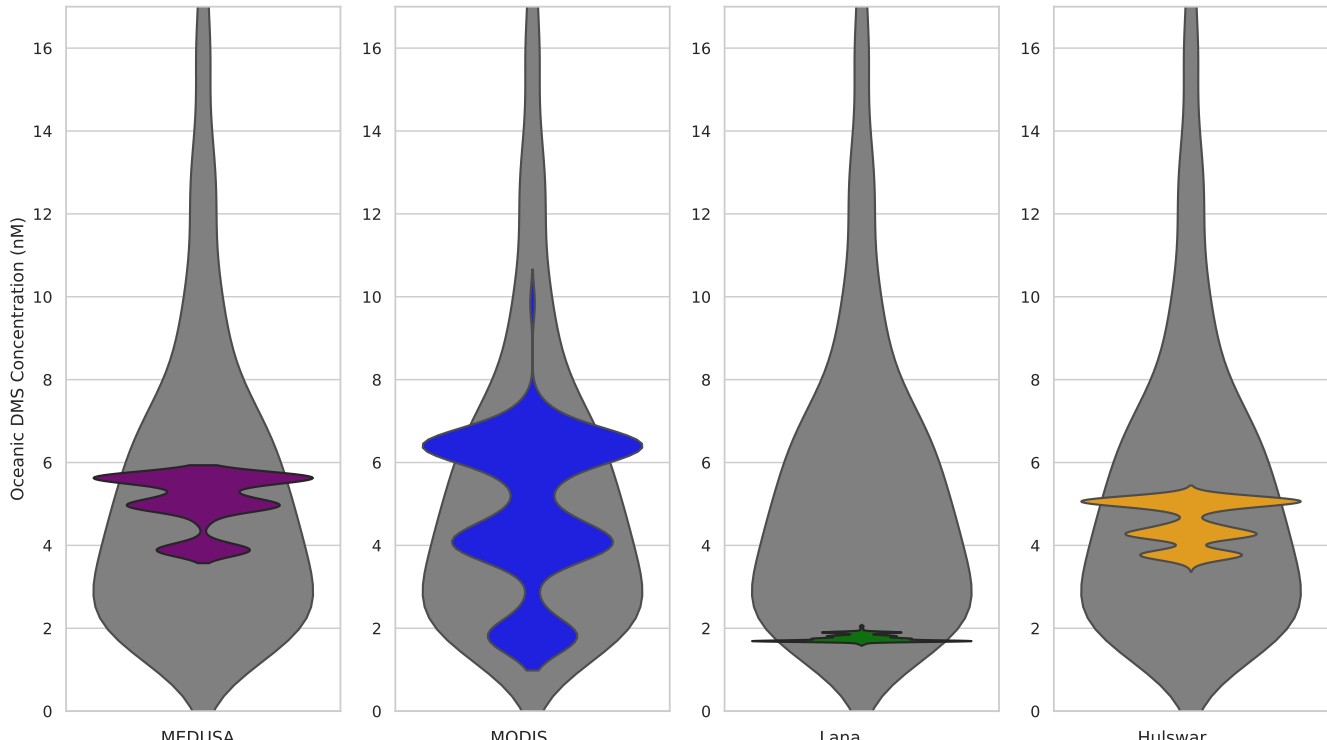

**Figure 4.** Same as Figure 3, but for the SOAP 2012 voyage (Feb to March 2012, 42–47 °S, 172–180 °E).

SOAP, where MODIS-DMS follows the concentrations through space and time better than the other datasets. For example, when SOAP measures its lowest oceanic DMS concentrations following the voyage, MODIS-DMS also simulates its lowest concentrations. Additionally, when SOAP observations are at their highest concentrations (25 nM), MODIS-DMS displays its highest concentration (over 10 nM). See Figure B1 for simulated comparisons of DMS emission to SOAP.

Lana, Hulswar, and MEDUSA fail to represent high biological variability in the Chatham Rise region of the Southern Ocean, as confirmed via comparison with TAN1802 between 45 °S to 60 °S. MODIS-DMS does not capture the heightened concentrations from SOAP or TAN1802, but it aligns more closely with the observations than the climatologies. This is likely due to the MODIS-DMS simulations using the chl-$a$ data during the period of each voyage, and nudging model conditions to similar conditions. From this, implementing the inclusion of satellite chl-$a$ in oceanic DMS calculations improves the accuracy of DMS distribution in lower latitudes.

The Anderson et al. (2001) oceanic DMS parameterization assumes chl-$a$ has a central role in forming oceanic DMS. The known global correlation between chl-$a$ and oceanic DMS, described by the coefficient of determination ($r^2$), is between 0.11 to 0.818, where higher latitudes tend to have higher $r^2$ values (Uhlig et al., 2019; Townsend and Keller, 1996; Tison et al., 2010; Matrai et al., 1993). The Anderson et al. (2001) parameterization used in MODIS-DMS, has a strong $r^2$ value of 0.75 in the Southern Ocean, validating this parameterization for simulating oceanic DMS. The Anderson et al. (2001) parameterization





using satellite chl-$a$ provides a better representation of austral summer oceanic DMS conditions within the Southern Ocean compared with the Anderson et al. (2001) MEDUSA configuration.

Chl-$a$ is used to calculate oceanic DMS within half the Earth System Models with interactive biogeochemistry participating in CMIP6 (Bock et al., 2021). CNRM-ESM2-1 uses a comprehensive prognostic approach that considers grazing by zooplankton and DMSP, rather than chl-$a$. However, this is very difficult to validate due to the lack of widespread data availability. Here we suggest that a realistic biological proxy, such as chl-$a$, is useful to construct an oceanic DMS dataset. An oceanic DMS algorithm developed by Galí et al. (2018) includes sea-surface temperature, chl-$a$, photosynthetically active radiation, and the mixed layer depth, where oceanic DMS has a general overestimation along coastal regions (Galí et al., 2019; Hayashida et al., 2020). Galí et al. (2018) also produced a time series of oceanic DMS over parts of the Northern Hemisphere, finding similar inter-annual variability using chl-$a$ satellite data. We concur with Galí et al. (2018) that a move beyond classical climatologies is an important step in developing future climate models. We suggest using temporally varying input instead of climatology to allow the capture inter-annual variability over the Southern Ocean, particularly from ENSO events and biologically productive years. One such way to achieve this for future projections would be through a stochastic approach of capturing all chl-$a$ years from the satellite (e.g. SeaWiFS and MODIS-aqua) archive, including high biological productivity years, such as 2010 and 2020, or low productivity such as in 2015 (Figure A1).

### 3.2 DMS Flux

Having established that the MODIS-DMS data set produces simulated oceanic DMS in good agreement with observations (Figure 3), we now test the sensitivity of atmospheric DMS to a suite of sea-to-air transfer functions for different oceanic DMS sources. Figure 5 shows the DMS flux during austral summer across all simulations integrated over the Southern Ocean region (40 to 60 °S), which ranges, on average, between 2.9 to 7.3 TgS Yr$^{-1}$. The spread of the mean fluxes across all eight simulations is 153%, which is greater than the difference between all the oceanic DMS inputs, a 107% spread in mean oceanic DMS concentration. The lowest CoV value within both oceanic DMS and DMS emissions are found in the MODIS-DMS simulations, specifically, the Blomquist et al. (2017) parameterization (MODIS$_{B17}$) with a mean of $2.9 \pm 0.84$ TgS Yr$^{-1}$. The upper range of simulated DMS flux, $7.3 \pm 1.8$ TgS Yr$^{-1}$, comes from the W14 quadratic formula used with the Lana DMS climatology (Lana$_{W14}$).

LM86 has a higher transfer velocity than B17 for wind speeds greater than 7.5 m s$^{-1}$ (Figure 1). The Southern Ocean has the highest wind speeds over any ocean region, with wind speeds very frequently above 7.5 m s$^{-1}$ (Bracegirdle et al., 2020), therefore our simulations show Liss and Merlivat (1986) flux produces 14% more emissions of DMS than Blomquist et al. (2017) (Figure 1). Lana is widely used by climate models (e.g. Sellar et al., 2019; Horowitz et al., 2020; Bhatti et al., 2022). Implementing a DMS flux based on DMS observations within this climatology (Lana$_{B17}$) results in a $4.86 \pm 1.67$ TgS Yr$^{-1}$ flux, which is within the range of all the simulations (Figure 5).

For simulations using the same LM86 sea-to-air flux parameterizations, but different oceanic DMS sources, the spread of all means is 112% (3.3 to 6.9 TgS Yr$^{-1}$). The means derived from different DMS flux parameterizations (LM86, B17, and W14) within MODIS-DMS and Lana are spread between 51% to 62%. The choice of the oceanic DMS source is therefore



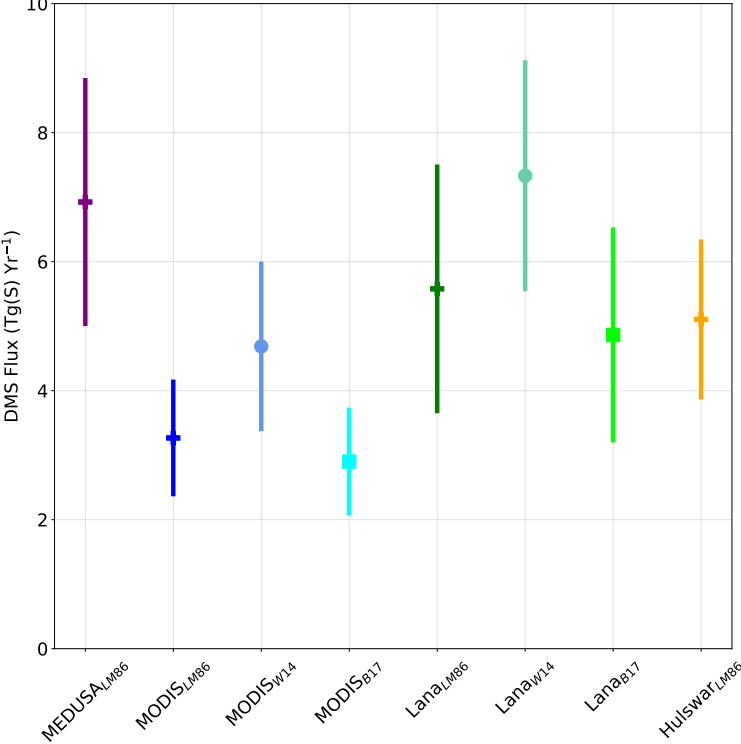

**Figure 5.** Summertime (December – February) Southern Ocean sulfur emissions in Tg Year$^{-1}$ in all model simulations performed. The error bars represent the spatial and temporal standard deviation. The different colors represent different oceanic DMS climatologies (Purple: MEDUSA ((Sellar et al., 2019; Anderson et al., 2001), Green: (Lana et al., 2011) and Orange: Hulswar ((Hulswar et al., 2022), and time series (Blue: derived from MODIS-DMS chl-$a$) used in this work. + marker represents simulations performed with the Liss and Merlivat (1986) sea-to-air flux, the dot marker represents Wanninkhof (2014), and the square marker represents Blomquist et al. (2017).

more important than the choice of DMS emission flux. Changing oceanic DMS within the model produces a larger impact on the resultant atmospheric DMS than the flux parameterization used. The emission of DMS from the ocean, over the Southern Ocean, results in a slightly higher spread between all the simulations of 6%. For a given oceanic DMS source, the quadratic formula of W14 produces around 33% more DMS emissions than the linear formulas of LM86 and B17.

The selection of a DMS flux parameterization has a large impact on the emissions of oceanic DMS. The W14 parameterization generates excessive DMS emissions, accurately representing the highest 10% (see Figure C1) but overestimating the rest of the distribution. Using W14, or similar quadratic fluxes (such as Nightingale et al. (2000)) within climate models for DMS emissions could therefore result in an overproduction of DMS. See Figure D1 for a visual overview of DMS across all simulations.

Table 3 presents simulated daily Southern Ocean DMS fluxes during the austral summer. The total annual DMS emissions which occur during DJF are presented as a percentage and as a daily flux. DMS emissions during austral summer make up





**Table 3.** Mean daily DMS flux over the Southern Ocean during the austral summer period for each simulation. The percentage shows the proportions of the total annual DMS flux which occurs during the summer months. Additionally, the total DJF flux shows the mean daily DMS flux ($\mu$mol m$^{-2}$ d$^{-1}$) and standard deviations. The last row outlines the overall DMS flux volume, simulated above the 2.5 $\mu$mol m$^{-2}$ d$^{-1}$ threshold for aerosol nucleation to occur from DMS emissions, as a percentage.

| | MEDUSA$_{LM86}$ | MODIS$_{LM86}$ | MODIS$_{W14}$ | MODIS$_{B17}$ | Lana$_{LM86}$ | Lana$_{W14}$ | Lana$_{B17}$ | Hulswar$_{LM86}$ |
|---|---|---|---|---|---|---|---|---|
| Total DJF % | 38.1 | 31.8 | 31.7 | 33 | 44.9 | 40 | 46 | 40.3 |
| Total DJF ($\mu$mol m$^{-2}$ d$^{-1}$) | 31.8 ± 5.9 | 15.3± 2.7 | 21.6 ± 3.8 | 13.3 ± 2.6 | 26.1 ± 4.6 | 34.4 ± 5.5 | 22.3 ± 3.9 | 24 ± 3.7 |
| Total DJF % above 2.5 $\mu$mol m$^{-2}$ d$^{-1}$ | 88 | 61 | 73.5 | 52 | 62 | 76.8 | 56 | 66 |

32-46% of the annual flux, substantially lower than the 72% reported by Webb et al. (2019). However, Webb et al. (2019) measured DMS in Ryder Bay, near the Antarctic Peninsula (67.54 °S, 68.35 °W), an area known for high levels of sea-ice melt and high DMS emissions during DJF. The daily mean flux from our simulations during DJF is 22.2 ± 5.13 $\mu$mol m$^{-2}$ d$^{-1}$. Compared with the 22.7 $\mu$mol m$^{-2}$ d$^{-1}$ flux reported by Webb et al. (2019) for Ryder Bay during the summer, our simulations therefore represent fluxes within the expected range. Additionally, a 2018 Southern Ocean voyage during February calculated a mean daily flux to be between 2.6 ± 3.5 $\mu$mol m$^{-2}$ d$^{-1}$ within the open ocean (Zhang et al., 2020). Tracking this voyage through space and time with our simulations shows fluxes varied between 2.7 $\mu$mol m$^{-2}$ d$^{-1}$ from MODIS$_{B17}$ to 8.9 $\mu$mol m$^{-2}$ d$^{-1}$ in Lana$_{W14}$. Shon et al. (2001) estimated the daily flux between 40 °S to 55 °S around early December to be 2.6 ± 1.8 $\mu$mol m$^{-2}$ d$^{-1}$. Only MODIS$_{B17}$ coincides with these daily fluxes across this latitudinal region. Our MODIS-DMS simulations using a linear flux parameterisation (LM86 and B17) also align with the 12 ± 15 $\mu$mol m$^{-2}$ d$^{-1}$ measured by Marandino et al. (2009) and the 2.8 $\mu$mol m$^{-2}$ d$^{-1}$ measured by Lee et al. (2010) in the Southern Ocean.

Pandis et al. (1994) estimates that for aerosol nucleation to occur from DMS emissions, the flux must be above 2.5 $\mu$mol m$^{-2}$ d$^{-1}$. Our simulations show that DMS emissions are above this threshold between 52% (MODIS$_{B17}$) and 88% of the time (MEDUSA$_{LM86}$) during summertime. This range agrees with Webb et al. (2019), who measured the flux to be over this threshold around 63% of the year. MODIS$_{LM86}$ (at 61%) and Lana$_{LM86}$ (at 62%) compare best to the observed value, although Webb et al. (2019) is likely positively skewed based on their location in an area characterised by large DMS emissions. As the UKESM1 underestimates AOD during austral summer (Mulcahy et al., 2020), MEDUSA$_{LM86}$ also produces the highest daily DMS flux over the Southern Ocean, suggesting a bias may be present during the formation of aerosols. This may be a result of the chemistry scheme used in the formation of aerosols as a compensating bias, which will be addressed in future work.

Many CMIP6 models use the quadratic sea-to-air flux parameterization detailed in Wanninkhof (2014) to calculate DMS emissions (e.g. Salzmann et al., 2022; Seland et al., 2019; Neubauer et al., 2019; Tatebe and Watanabe, 2018; Wu et al., 2018), however, recent literature suggests that DMS has a linear relationship with wind speed (e.g. Blomquist et al., 2017; Goddijn-Murphy et al., 2016; Bell et al., 2013; Zavarsky et al., 2018; Vlahos and Monahan, 2009; Bell et al., 2015). We show that linear DMS emissions may not represent the upper ranges of DMS flux as well as quadratic flux emissions, where wind and oceanic DMS concentrations are high. Extreme concentrations of oceanic DMS can result in very high emissions. Lana$_{W14}$ simulates these higher concentrations similar to the higher fluxes from TAN1802 (Figure C1) but result in an overestimation for the lower





emissions of the distribution. MEDUSA$_{LM86}$ emits DMS similarly to the quadratic formulation of Lana$_{W14}$, within the higher

ranges of emissions. Therefore, using a formula developed specifically for DMS, such as Blomquist et al. (2017) may generally better represent DMS emissions in the UKESM1. Additionally, simulations of atmospheric DMS with UKESM1 are improved when using observed chl-$a$ concentrations to calculate historical oceanic DMS.

### 3.3 Atmospheric DMS

We now evaluate atmospheric DMS in our sensitivity simulations. Figure 6 compares all simulated atmospheric DMS con-

centrations to observational data averaged across the Southern Ocean during austral summer. The observational data shown in Figure 6 was collated from three observational stations (Cape Grim, Amsterdam Island, and King Sejong Station) and three Southern Ocean voyages (SOAP, SOIREE, and ANDREXII), with an average and a standard deviation (spatial and temporal) summertime atmospheric DMS concentration of $185 \pm 129$ ppt (Smith et al., 2018; Wohl et al., 2020; Boyd and Law, 2001). The mean atmospheric DMS across all simulations is $276 \pm 174$ ppt, and is within the range of the observations. In addition,

when using the DMS source in best agreement with oceanic observations (MODIS-DMS) combined with a linear DMS flux parameterisation (LM86 and B17), the atmospheric concentration mean is consistent with the observational mean, averaging $164 \pm 132$ ppt. Along the Peruvian coastline, Zhao et al. (2022) measured atmospheric DMS concentrations at $145 \pm 95$ ppt, which also aligns well with the linear MODIS-DMS simulations. During the summer months, mean atmospheric DMS concentrations of 119 ppt (measured at 64.8 °S, 64 °W) by Berresheim et al. (1998) and 114 ppt (measured at 75.4 °S, 26.2 °W)

by Read et al. (2008) align best with the MODIS-DMS simulations. Additionally, Lee et al. (2010) measured a mean of 61 ppt over the same high latitudes in February. However, there are also disagreements between observations and MODIS-DMS simulations with linear fluxes, as a voyage tracking along the Eastern South Pacific Ocean during January 2000 measured 340 $\pm$ 370 ppt (Marandino et al., 2009), which is consistent with Lana$_{LM86}$, but not as high as Lana$_{W14}$. The variability from Amsterdam Island measurements is much higher than that of the simulations.

These measurements highlight the high variability of atmospheric DMS during austral summer over the Southern Ocean. Berresheim (1987) measured 106 ppt over the Drake passage during March and April, representing the lower end of our simulated DMS mixing ratio. All measurements during summer show very high variability, with lower values seen in higher latitudes. MODIS$_{B17}$ does a better job of representing atmospheric DMS compared to simulations from other models like MEDUSA, Lana, and Hulswar when compared to observations.

So far we have focused on DMS, which is an important biogenic marine aerosol precursor. However, the development of the MODIS-DMS data set has implications for primary marine organic aerosol (PMOA), whose production is influenced by chl-$a$ concentration in UKESM1 (Mulcahy et al., 2020). PMOA are organic detritus or compounds that are emitted to the atmosphere when bubbles burst as waves break (Gantt et al., 2012, 2011). A parametrisation of PMOA has been implemented in the UKESM1 in Mulcahy et al. (2020) based on the parametrisation developed by Gantt et al. (2011), where PMOA is a

function of wind speed, sea salt dry diameter, and surface chl-$a$. As identified by Mulcahy et al. (2020), the MEDUSA chl-$a$ bias is carried through to PMOA which results in a Southern Ocean distribution similar to that of oceanic DMS.





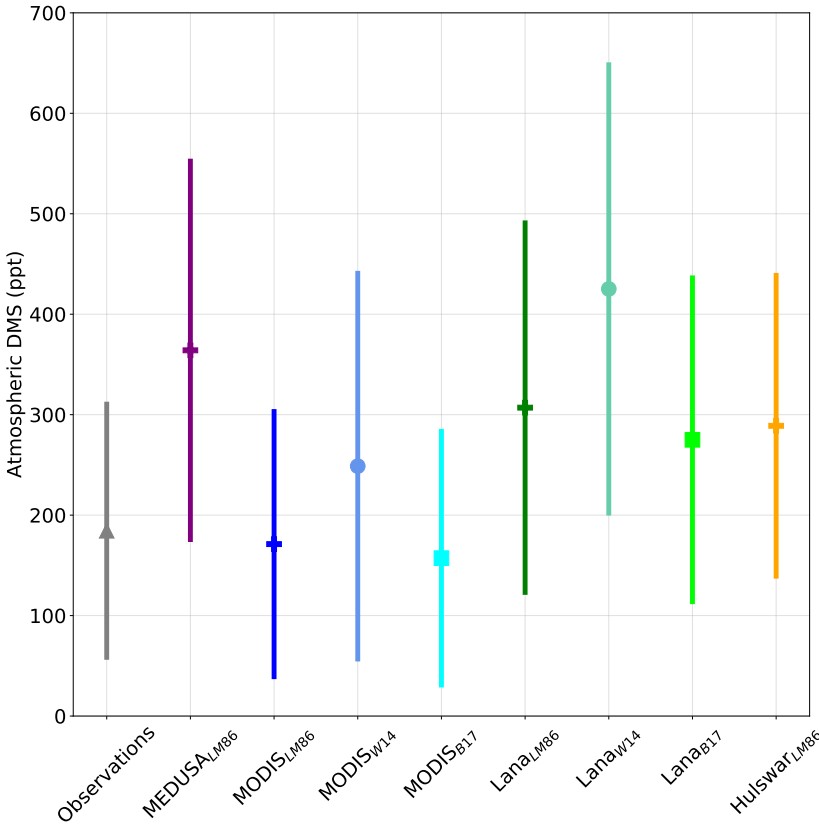

**Figure 6.** DJF averaged atmospheric concentration (ppt) for the nine simulations. The observations represent a summertime average across Cape Grim, Amsterdam Island, and King Sejong Station and three Southern Ocean voyages (SOAP, SOIREE, and ANDREXII). The error bars represent the standard deviation through time and space.

PMOA is the dominant source of ice nucleating particles over the Southern Ocean (Vergara-Temprado et al., 2018; Zhao et al., 2021) which may reduce the downwelling shortwave radiation bias (Schuddeboom and McDonald, 2021; Fan et al., 2011; Fiddes et al., 2022). This shortwave bias may have links to a deficit in supercooled liquid over the Southern Ocean within CMIP6 (Fan et al., 2011). Our analysis shows that upon implementing chl-$a$ measurements derived into the parametrisation (Figure 7), PMOA emissions over the Southern Ocean increase substantially during summer ( 81%). Although PMOA is in the preliminary stages of development within the UKESM1, our results increase in ice nucleating particles which could improve PMOA-driven cloud formation processes over the Southern Ocean. This will be investigated in future work.




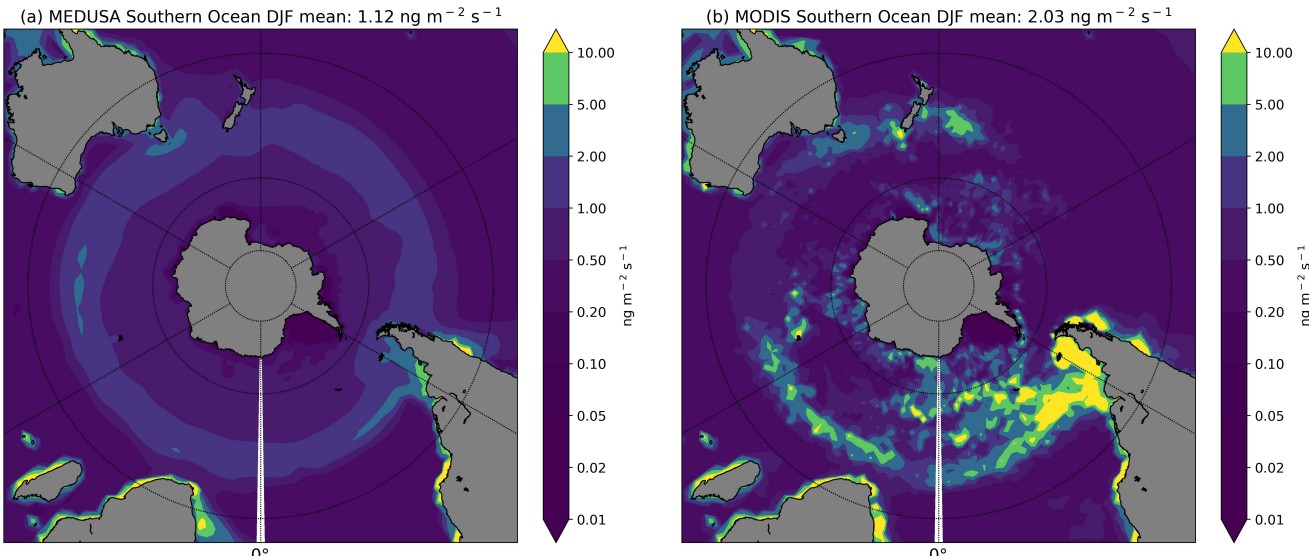

**Figure 7.** Emissions of primary marine organic aerosol for (a) MEDUSA and for the (b) MODIS-DMS simulation integrated over the summer period (DJF). The mean value is area weighted across the Southern Ocean DJF.

# 4   Conclusions

The concentration and distribution of atmospheric DMS is highly uncertain over the Southern Ocean, in part due to the lack of observational data and lack of understanding of oceanic DMS and DMS emissions. We examine the key processes and relationships involved in the emissions of DMS and the production of atmospheric DMS. We also provide an overview of different oceanic DMS climatologies and calculate an oceanic DMS time series using chl-$a$ satellite data. We then used three different oceanic DMS climatologies (MEDUSA, Lana, and Hulswar) in the UKESM1-AMIP model. We also constructed

a competing oceanic DMS spatially distributed time series, using satellite chl-$a$ data from 2009 to 2018. By using nudged simulations, we can more accurately examine the drivers of the change between oceanic DMS and atmospheric DMS via closer comparisons with observations. Across all four oceanic DMS datasets, we used the sea-to-air parameterization proposed by Liss and Merlivat (1986). We incorporated a quadratic parameterization from Wanninkhof (2014) in the MODIS-DMS and Lana oceanic DMS source to align with current flux estimations in the literature and other Earth System Models. Moreover,

we tested a formula based on DMS observations from Blomquist et al. (2017) within the MODIS-DMS and Lana simulations.

MODIS-DMS suggests that large areas of open water in the Southern Ocean have lower oceanic DMS concentrations compared with MEDUSA, Lana, and Hulswar. Our study finds that climatologies based on observations show fewer distinct features in oceanic DMS concentrations. On average, all four oceanic DMS datasets have a summertime mean of $3.7 \pm 1.2$ nM within the Southern Ocean (40 °S to 60 °S). MODIS-DMS oceanic DMS shows significant differences in the Southern



Ocean between coastal areas and the open ocean, where coastal regions contain enhanced oceanic DMS. By incorporating a time series based on proxies of real-world biological data, we demonstrate that annual chl-$a$ fluctuations can influence oceanic DMS and impact emissions. This highlights the importance of capturing high levels of biological activity within oceanic DMS over time.

We find that atmospheric DMS is more sensitive to changes in oceanic DMS than the range of flux parameterizations used 400 in this study. Using different oceanic DMS concentrations with the same sea-to-air parameterization results in a 112% spread across the means within the DMS emissions. In contrast, changing just the DMS flux parameterization results in a spread of 50-60%. Additionally, atmospheric DMS concentrations are more sensitive to changes in oceanic DMS concentrations than DMS emissions. The mean emissions between all eight simulations have a spread of 153%, smaller than the spread across the atmospheric concentration of 171%. Changing either the oceanic DMS or DMS flux parameterization has considerable effects 405 on atmospheric DMS concentrations and emissions, thus requiring careful thought about implementation in future simulations.

We recommend moving away from the commonly used W14 quadratic sea-to-air flux parameterization in CMIP6 models for DMS and instead consider more up-to-date relationships developed specifically for DMS. The Wanninkhof (2014) quadratic DMS parameterization has a 33% larger influence on DMS emissions than that of Liss and Merlivat (1986) and Blomquist et al. (2017). Additionally, all simulations have a summertime Southern Ocean flux of $22.2 \pm 5$ $\mu$mol m$^{-2}$ d$^{-1}$, with linear 410 flux parameterizations aligning better with observations than the quadratic flux. Furthermore, we found that using a linear flux parameterization, B17, and LM86, within MODIS-DMS aligned the atmospheric DMS ($164 \pm 132$ ppt) much closer to the observations ($185 \pm 129$ ppt). All simulations have a Southern Ocean DJF mean of $276 \pm 174$ ppt, within one standard deviation.

The use of climatologies within climate models to represent oceanic DMS does not represent realistic distributions within 415 the Southern Ocean as shown by comparisons to voyages. Climatologies should be replaced by spatially distributed time series to represent inter-annual variability of oceanic DMS. The time series would benefit from using a wide-spread readily available dataset that best represents a realistic spatial distribution of oceanic DMS. Given the current data availability, using chl-$a$ data from the MODIS-aqua satellite is a viable option. The oceanic DMS within the UKESM1 (MEDUSA) is positively biased (e.g. Bock et al., 2021; Mulcahy et al., 2020; Yool et al., 2021), and is in need of further development. In future work, 420 when developing sulfate chemistry, we recommend using the LM86 or B17 flux parameterization along with either Lana or MODIS-DMS oceanic DMS concentrations, to capture a more realistic DMS cycle in the Southern Ocean.

*Author contributions.* Author contributions. YAB implemented model developments, performed model simulations and wrote the manuscript with assistance from all co-authors. LER, AJM and AJS assisted with the experimental design and the evaluation of the model compared with the observational dataset and sensitivity analysis. ATA advised on DMS chemistry and aerosols over the Southern Ocean. CH provided 425 assistance for lodging DMS emissions into the UKESM1. JW and EB provided technical expertise in running model simulations. AUV and LER advised on PMOA in the UKESM1 over the Southern Ocean.



*Competing interests.* no competing interests are present

*Acknowledgements.* This research was supported by the Deep South National Science Challenge (Grant Nos. C01X141E2 and C01X1901) and the UK Met Office for the use of the MetUM. We also wish to acknowledge the contribution of New Zealand eScience Infrastructure (NeSI) high-performance computing facilities to the results of this research. New Zealand's national facilities are provided by NeSI and funded jointly by NeSI's collaborator institutions and through the Ministry of Business, Innovation and Employment's Research Infrastructure programme (https://www.nesi.org.nz/, last access: 06 April 2023). We acknowledge the Cape Grim Science Program for the provision of DMS data from Cape Grim. The Cape Grim Science Program is a collaboration between the Australian Bureau of Meteorology and the CSIRO Australia.



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



**Appendix A**





**Figure A1.** Timeseries of each oceanic DMS dataset input into the simulations during DJF in the Southern Ocean. chl-*a* is shown with a black line. The MODIS-DMS error bars represent the standard deviation across the Southern Ocean.





**Figure B1.** DMS emissions for each simulation tracking the SOAP voyage across each hour, through time and space as a violin plot. The $r^2$ value compares the monthly DMS flux vs the respective oceanic DMS. Spearman's rank compares the rankings of hourly flux simulation with the rankings of hourly TAN1802 data. Each simulation overlays the corresponding SOAP voyage flux, whereby both emissions are calculated with the same sea-to-air flux, winds, and Schmidt number.





**Figure C1.** Same as Figure C1, but for TAN 2018



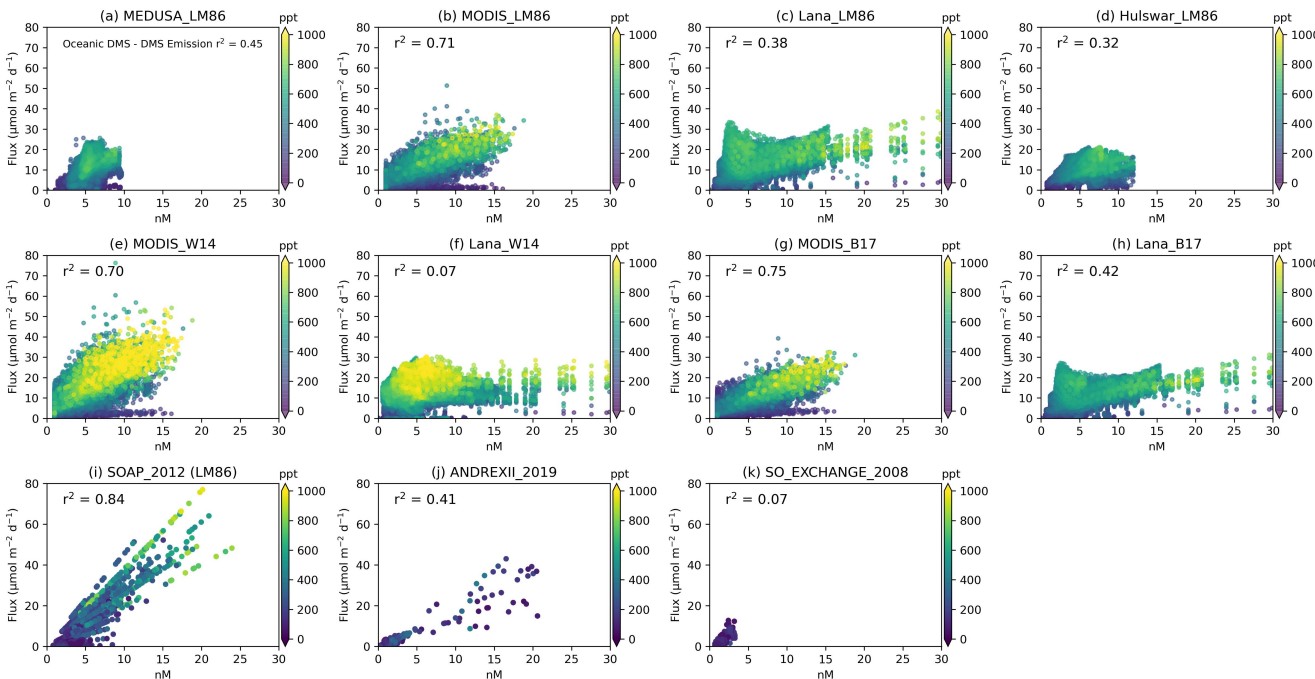

**Figure D1.** Scatter plot representing how DMS flux and atmospheric DMS mixing ratio respond to increasing oceanic DMS concentrations. (a-i) Each simulation presents 97920 data points within the DJF Southern Ocean. (j,k) The relationship between observational voyages during 2012 (SOAP using the Wanninkhof (2014) flux parameterization), 2019 (ANDREXII), and 2008 (SOExchange) is also presented. The $r^2$ value is shown on each plot to represent the coefficient of determination between the oceanic DMS and DMS emissions for each simulation and voyage.