# Peer review of "The sensitivity of Southern Ocean atmospheric dimethyl sulfide to modelled sources and emissions"

_EGUsphere, 2023_

## Author Response (AR1)

We appreciate the reviewers for their constructive comments and suggestion, which has greatly improved the quality of our manuscript. Below we respond to each comment from each reviewer. Reviewer comments are shown in black, our response is in *red italics*, and revised text is in blue. Also available in a tracked change document.

**Response to Referee #1**

**Investigation of inter-annual variability**: The authors conclude that interannual variability is important (e.g. L18, L416), but this is not demonstrated well in the study. Figure A1 is the only result where the interannual variability can be discussed, bit it gives an impression that the interannual variability is not important (the year-to-year variation of the mean DMS concentration is subtle). To assess the impact of representing interannual variability in the DMS source, the authors could do an additional simulation in which MODIS-DMS does not change from year to year. By comparing this simulation with one of the MODIS simulations in Table 2, the impact can be quantified and its importance can be justified.

*We thank the reviewer for this comment. We have performed such a simulation which is similar to the MODIS$_{B17}$ simulation, but the MODIS-DMS time-series used in MODIS$_{B17}$ is replaced with a climatology calculated from MODIS-derived seawater DMS concentrations over the 10-year period. We have provided some additions to the methodology section to describe this simulation and discuss interannual variability in the results and discussion section.*

**2.3 DMS Sea-to-Air Flux:**

To assess the inter-annual variability of DMS emissions and atmospheric DMS concentrations, we performed an additional 10-year simulation, MODIS$_{B17}$CLIM. While MODIS$_{B17}$ used a 10-year time series of oceanic DMS derived from MODIS chlorophyll-a data, MODIS$_{B17}$CLIM used a climatology calculated from monthly-mean data for the 10-year MODIS$_{B17}$ time series.

**3.4 Effects from Inter-annual Variability and Spatial Variability:**

To assess the impact of interannual variability in oceanic DMS on simulated atmospheric DMS, we compare the MODIS$_{B17}$ simulation with MODIS$_{B17}$CLIM, which used a climatology of oceanic DMS calculated from the MODIS-DMS data set (Figure 7). Both simulations are similar ($R^2$ = 0.92) in terms of interannual variability across the Southern Ocean as a whole. (Figure 7c). Rolling means are presented in Figure A1b, c. While there are small differences in Southern Ocean atmospheric DMS between the simulations, the overwhelming similarities between Figure 7a and b suggest that an oceanic DMS climatology results in similar interannual variability in the atmospheric DMS PDF suggesting that oceanic DMS is not a strong driver of interannual variability in atmospheric DMS. This result is in contrast to that of Galí et al. (2018) who used a different algorithm for producing oceanic DMS. This difference may be due to our use of the Anderson et al. (2001) algorithm, which is known to produce limited variability (Belviso et al., 2004; Bock et al., 2021).

To assess the impact of spatial variability in oceanic DMS on simulated atmospheric DMS, we compare simulations performed using the MEDUSA and MODIS-DMS data sets (with low and high spatial variability in oceanic DMS, respectively) in Figure 8. Larger variability in the MODIS-DMS oceanic DMS source leads to larger variability in simulated atmospheric DMS, compared with the MEDUSA simulations. The spatial CoV from MEDUSA$_{LM86}$ is 45% lower than MODIS$_{LM86}$, showing greater spatial variability from MODIS-derived chl-a. The oceanic DMS signal to the atmosphere is strong but has high fluctuations from the wind.

[Figure]

Figure 7. Time series of the atmospheric DMS probability density function between (a) MODIS$_{B17}$ and (b) MODIS$_{B17}$CLIM from 2009 to 2018 summer over the entire Southern Ocean. (c) the difference between MODIS$_{B17}$ and MODIS$_{B17}$CLIM is also shown, with the $R^2$ shown between the two simulations.

[Figure]

*Figure A1. Timeseries of each oceanic DMS dataset input into the simulations during DJF in the Southern Ocean. (a) chl-a is shown with a black line. The MODIS-DMS error bars represent the standard deviation across the Southern Ocean. (b) comparison of the DMS emissions between MODISB17 with the MODISB17CLIM. (c) as (b) but showing the surface atmospheric DMS concentrations. The legend highlights the standard deviation in annual flux, along with the R2 value between the two simulations. (bottom) as for the (middle) but showing atmospheric DMS concentrations*

The choice of oceanic DMS parameterization: The authors chose Anderson et al. (2001) for deriving DMS from chlorophyll-a, while there are other algorithms to derive DMS from chlorophyll-a and other environmental variables developed since 2001.This choice needs to be explained/justified in the manuscript.

*Anderson et al. (2001) is the native oceanic DMS parameterization within UKESM1. While it is not the most up-to-date oceanic DMS parameterization (e.g. Bock et al., 2021; Simo and Dachs., 2002; Vallina and Simo., 2007), the aim here is simply to investigate the effect of variability in atmospheric DMS*

*We have added justification for our approach as described:*

**2.2 Oceanic DMS (Methods):**

In UKESM1, oceanic DMS concentrations are calculated using a diagnostic method from Anderson et al. (2001), using surface daily shortwave radiation (J), dissolved inorganic nitrogen (Q), and chl-a (C):

$$Oceanic\ DMS\ =\ a, for\ log(CJQ)\ \leq\ s$$

( 1 )

$$Oceanic\ DM\ S\ =\ b[log(CJQ)\ -\ s]\ +\ 1, for\ log(CJQ)\ >\ s$$

(2)

The parameter values are a=1, b=8, and s=1.56, as described by Sellar et al. (2019). Q, chl-a, and J are averaged from CMIP6 for the MEDUSA climatology. The Anderson et al. (2001) parameterization produces positive biases in DMS over the Southern Ocean within MEDUSA (Bock et al., 2021) due to the set minimum oceanic concentration of 1, which leads to large average DMS concentrations (Yool et al., 2021; Bock et al., 2021). Recent research suggests that chl-a may not be an appropriate proxy for oceanic DMS (Uhlig et al., 2019; Bell et al., 2021), and future work will explore alternative methods for calculating oceanic DMS within UKESM1. Nonetheless, chl-a is widely used by CMIP6-era models to calculate oceanic DMS, and we explore here whether using an observationally derived chl-a concentration field leads to changes in the spatial and temporal variability of atmospheric DMS. Monthly-mean chl-a concentrations from the Moderate Resolution Imaging Spectroradiometer (MODIS)-aqua satellite instrument were used to construct a time series of oceanic DMS between 2009–2018 (Table 1; Hu et al., 2019; O'Reilly and Werdell, 2019). This time series, which we term the 'MODIS-DMS' data set, is calculated offline using the same diagnostic parameterization as Equations 1 and 2. The J and Q used to calculate MODIS-DMS remain the same as MEDUSA. Through this, we capture spatial and interannual chl-a variability, indicating biological productivity. Bi-linear interpolation is used to fill in small gaps (around 1% for monthly averages) of spatial chl-a data. Oceanic

DMS concentrations are masked where they coincide within the sea-ice zone from HadISST.

*In addition to this, we also further justified the use of using chl-a to calculate oceanic DMS in 3.1 Oceanic DMS, as outlined below:*

The Anderson et al. (2001) parameterization assumes chl-a is central to oceanic DMS formation. Previous correlations between chl-a and oceanic DMS, given by the coefficient of determination ($R^2$), range globally from 0.11 to 0.93, with higher latitudes having increased $R^2$ values due to factors like nutrient availability and prolonged summer daylight, coupled with heightened wind speeds (Uhlig et al., 2019; Townsend and Keller, 1996; Tison et al., 2010; Matrai et al., 1993). Gros et al. (2023) estimated an $R^2$ of 0.93 towards sea ice latitudes, while Bell et al. (2021) found chl-a explains just 15% of oceanic DMS variability. Using the Anderson et al. (2001) parameterization in MODIS-DMS, we determined a large $R^2$ of 0.75 in the Southern Ocean. While associating chl-a with oceanic DMS has discrepancies (Gros et al., 2023; Bell et al., 2021), we show that using Anderson et al. (2001) with satellite chl-a data better represents Southern Ocean summertime DMS compared with the MEDUSA configuration.

Investigation of spatio-temporal variability: The authors introduce the importance of spatio-temporal variability (L57) of DMS in the Southern Ocean, but the results are mostly discussed for spatially and temporally averaged values. Furthermore, some model results are averaged over the entire Southern Ocean, which are compared to the observations that are representative of specific regions/seasons (e.g. L312, L318, L320, L323, L352). While apple-to-apple comparison may not be possible, there are places for improvement. For example, the spatial maps of the simulated DMS flux and atmospheric DMS (such as done for oceanic DMS concentration as in Fig 2a-d) can be added to better compare with the observations from specific regions. These figures will also demonstrate the importance of spatial variability.

*We thank the reviewer for this suggestion. To provide a better outline of the spatio-temporal variability, we have included two new figures in the main text showing the spatial variability for the atmospheric DMS across four individual days. This has been implemented into subsection 3.4 Effects from Inter-annual Variability and Spatial Variability. We also direct the reviewer to Figure A4, which shows the daily spatial and temporal distribution of all simulated emissions as box plots. Based on the chlorophyll inter-annual variability shown in Figure A1, we compare the DJF averages for 2 years of peaks, in 2009 and 2010, and two years of declines in 2013, and 2015 (Figure 8).*

*Additional text is provided in the manuscript to support these new figures and repeated below:*

To assess the impact of spatial variability in oceanic on simulated atmospheric DMS, we compare simulations performed using the MEDUSA and MODIS-DMS data sets (with low and high spatial variability in oceanic DMS, respectively) in Figure 8. Larger variability in the MODIS-DMS oceanic DMS source leads to larger variability in simulated atmospheric DMS, compared with the MEDUSA simulations. The spatial CoV from MEDUSA is 45% lower than MODIS$_{LM86}$, showing greater spatial variability from MODIS-derived chl-a. The oceanic DMS signal in the atmosphere is strong but includes large fluctuations the wind.

[Figure]

Figure 8. Atmospheric DMS concentrations comparing (a - d) MEDUSALM 86 with (e - h) MODISLM 86 across four of the same summertime days (15$^{th}$ December) in (a, e) 2009, (b, f) 2010, (c, g) 2013, (d, h) 2015. The area-weighted Southern Ocean mean is shown below each plot.

[Figure]

*Figure A4. DMS emissions for all simulations over the 10 years using daily data in the summertime (DJF) over the Southern Ocean.*

writing style: Section 5 seems like a combination of discussion and conclusions, so I suggest to change the section title to "Discussion and conclusions". However, I also noted in my specific comments that there are paragraphs in the Results section that I feel belong to Discussion.

*We thank the reviewer for bringing this to our attention. All the comments and results discussed in the conclusion section have now already been stated within the results section. To better label this results section, we have renamed it: Results and Discussion. In addition to this, we have also minimized the discussion, restating just the key findings in the conclusion section.*

**Specific comments**

L3: Suggest to delete or rewrite this sentence. The authors did not assess how DMS emissions change when the chlorophyll-a distribution is altered. This would require an investigation of the relationship between the DMS emissions and the chlorophyll-a distribution.

*Deleted.*

L9: "The mean spread" of which quantity?

*Changed from*

The mean spread across all the simulations with different oceanic DMS datasets, but the same sea-to-air flux parameterizations, is 112% (3.3 to 6.9 TgS Yr−1).

*to*

In simulations with different seawater DMS data sets but the same sea-to-air flux parameterization, Southern Ocean summertime DMS varies by 112% (3.3 to 6.9 TgS Yr$^{-1}$). This is approximately twice as much as the simulations using the same seawater DMS data set but differing sea-to-air flux parameterizations, in which DMS varies by 50-60% (2.9 to 4.7 TgS Yr$^{-1}$).

L11: "The choice of" Is this conclusion correct? The previous sentences indicate that the choice of the oceanic DMS source (112%) has a larger influence on the atmospheric DMS than that of the flux parameterization (50-60%).

*Thank you for bringing this to our attention, we accidentally reversed the order. This has now been corrected to:*

The choice of oceanic DMS source has a larger influence on atmospheric DMS than the choice of DMS emission.

L12: Suggest to rewrite this sentence. It is not that linear relationships compare better to observations, but assuming/simulation with the linear dependence of the DMS flux on wind speed results in better representation of atmospheric DMS distribution.

*This sentence has been re-written as:*

Simulations testing different sea-to-air transfer velocity parameterizations show that simulating a linear dependence of DMS gas transfer velocity as a function of wind speed results in a more accurate representation of atmospheric DMS distributions than using quadratic relationships.

L15: Where is the evidence for the closest agreement? This could be clarified in "Discussion and conclusions".

*We have modified this sentence to align with our findings:*

Simulations using seawater DMS derived from satellite chlorophyll-a data show realistic spatiotemporal variability in DMS and when combined with recently developed transfer velocity parameterization for DMS, the model shows good agreement with atmospheric DMS observations.

L34: Suggest to replace "mechanisms, with varying focus" by "approaches that are dependent".

*done*

L40: Suggest to replace "approximate" by "prescribe".

done

L48: Suggest to repalce "and has" by "and this emission has".

done

L55: Older quadratic approaches are based on other gases? If so, this could be mentioned here to emphasize the issue.

Updated quadratic approaches, such as the Wanninkhof (2014) transfer velocity, based on $CO_2$ are currently used to simulate DMS emissions in many Earth System Models (Bock et al., 2021). We have updated the introduction section which now highlight the differences between the quadratic and linear approaches as such:

The flux of DMS from the ocean to the atmosphere depends on the gas transfer velocity, which in turn depends on the surface wind speed (e.g. Fairall et al., 2011). Many DMS flux parameterizations have been developed, but most use transfer velocities measured for gases other than DMS (Wanninkhof, 1992, 2014; Nightingale et al., 2000; Liss and Merlivat, 1986). Some studies, including Blomquist et al. (2017) and Yang et al. (2011), used DMS measurements to derive a relationship between wind speed and DMS. Depending on the solubility of the gas measured, gas transfer velocities typically have a linear or quadratic dependence on wind speed. Linear relationships best represent gases with intermediate solubilities, such as DMS (e.g. Blomquist et al., 2017; Goddijn-Murphy et al., 2016; Bell et al., 2015; Yang et al., 2011; Huebert et al., 2010), while quadratic equations are better suited for highly soluble gases like $CO_2$ (Wanninkhof, 2014; Nightingale et al., 2000; Wanninkhof, 1992).

L57: Should "observations" be "concentrations"? Observations are highly variable implies that some regions/seasons are observed more than the others. I am not sure if this is what the authors meant to state here.

Done. We meant concentrations

L59: Suggest to replace "real-world" by "remotely-sensed".

done

L61: Suggest to either remove "nudged to observation" or replace it by "nudged-to-observation".

Changed to nudged-to-observation

L68: "the relative importance" to what?

*To add clarity to this sentence, we have modified it from:*

We compare the DMS variability across the Southern Ocean during summer, improving our understanding of the relative importance of choosing the source (oceanic DMS) and emissions.

*To:*

We evaluate sea-to-air fluxes of DMS and oceanic and atmospheric DMS concentrations relative to station and ship-based observations.

L73: Suggest to move the description of MEDUSA later when the oceanic DMS dataset is introduced. The model used here is the atmopsheric-only configuration, so the oceanic component does not need to be introduced here.

*We have moved this description to the 'Oceanic DMS' section of the methodology.*

L85: In addition to the details provided in this section, I suggest to add information on boundary conditions such as sea surface temperature data used for these simulations that affect both the atmospheric circulation and DMS emissions (Eq 3).

*We have moved the description of the sea surface temperature from the sea-to-air flux to the initial descriptions of the model.*

Sea surface temperature and sea ice data from The Hadley Centre Global Sea Ice and Sea Surface Temperature were used (HadISST; Titchner and Rayner, 2014).

L86: Suggest to rewrite "biologically productive" to something else. "Biological" is a bit ambiguous, as it is broad. It could refer to microalgae to big animals that have different seasonality. For clarity, it may be better to refer to DMS instead of biology.

*We modified this paragraph to be more specific DMS as suggested, with a change from:*

All simulations in this study are 10 years long, spanning 2009 to 2018. We focus on the austral summer months (December-February; DJF) due to the summer being the most biologically productive season.

*to*

Simulations are 10 years long, spanning from 2009 to 2018. This period was chosen to coincide with the availability of recent DMS observations (Section 2.4).

L88: This paragraph belongs somewhere else, as it is not a model description.

*We have renamed this subsection:*

Model Configuration and Evaluation

L110: "modified" implies that the parameterization was modified, but my impression is that the authors only modified the chlorophyll dataset. If so, please rewrite this sentence.

*This sentence has been rewritten for clarity to state that the parameterization was not modified, but rather the chl-a from MEDUSA.*

Here, we have tested a modified version of it using the MODIS-aqua chl-*a* dataset.

*to*

Monthly-mean chl-a concentrations from the Moderate Resolution Imaging Spectroradiometer (MODIS)-aqua satellite instrument were used to construct a time series of oceanic DMS between 2009–2018 (Table 1; Hu et al., 2019; O'Reilly and Werdell, 2019).

L115: Suggest to define the acronym MODIS when it appears for the first time in the text instead of here.

*Done.*

L117: How are DMS sources and emissions from sea ice covered areas treated? Small gap of 1% probably excludes ice-covered grid cells. DMS emissions from marginal ice zone can be important (https://doi.org/10.1525/elementa.2020.00113).

*Similar to all oceanic DMS climatologies, where the model contains sea ice, oceanic DMS is not emitted. We make a note of this in the manuscript:*

Bi-linear interpolation is used to fill in small gaps (around 1% for monthly averages) of spatial chl-a data. Oceanic DMS concentrations are masked where they coincide within the sea-ice zone from HadISST.

L148: It is unclear how applying W14 and B17 parameterizations leads to testing the lower limits.

*This sentence has been updated:*

Using these different parameterizations provides an appropriate estimate for the spread of DMS emissions due to the upper and lower limits of DMS transfer velocity tested from in-situ DMS measurements (e.g. Goddijn-Murphy et al., 2016; Blomquist et al., 2017).

L158: Related to my earlier comment on L85, was HadISST used also for the boundary conditions for the atmospheric circulation simulations?

*Yes, all simulations used the same set of prescribed SSTs and sea ice concentrations. Only the oceanic DMS concentration and/or DMS flux changed.*

L172: Suggest to replace "validate" by "evaluate".

done

L201: Suggest to replace "includes chlorophyll-a" by "accounts for chlorophyll-a variations".

done

L209: Suggest to move "respectively" to "used, respectively)".

done

Fig 3: For readers not familiar with violin plots, the caption needs a brief description of what violin plots represent in general.

*Figure 3 caption now contains a brief description of violin plots, shown below.*

Violin plots depict data distribution and density. The width of each 'violin' corresponds to the frequency of data points within that value range, while the length indicates the range of values. The frequency axis, represented by the width, allows for an immediate visual comparison of how often particular ranges of values occur in each category. This offers a comprehensive view of both the distribution and frequency of data across different categories.

L249: Suggest to either delete "and would be expected to be highly biased" or clarify what it is meant by highly biased. Also for fair comparison, when was the TAN1802 data taken (before/during/after blooms)?

*SOAP (Bell et al., 2015) and TAN1802 (Kremser et al., 2021) have very different campaign objectives, and thus present distinctly different distributions of oceanic DMS. We therefore highlight the importance of the differences between the campaigns in more detail. From:*

SOAP voyage data represents oceanic DMS concentrations during phytoplankton bloom events, therefore the shape of the observed DMS distribution (Fig. 4) is quite different to the TAN1802 data (Fig. 3) and would be expected to be highly biased.

*to*

For the SOAP voyage, which targeted phytoplankton bloom events (42–47°S, 172–180°E), the measured DMS distribution is skewed toward higher concentrations compared with the TAN1802 voyage (Figure 4). In contrast, TAN1802 transected the Southern Ocean without specific focus on bloom activity, yielding a range of DMS concentrations. We consider that SOAP is still useful as it offers insights into extreme conditions not reflected in other data sets.

L265: Suggest to add a few words to briefly explain why higher correlations are expected at higher latitude.

This sentence has been modified:

The global correlation between chl-a and oceanic DMS, given by $R^2$, ranges from 0.11 to 0.93, with higher latitudes having increased $R^2$ due to factors like nutrient availability and prolonged summer daylight, coupled with heightened wind speeds (Uhlig et al., 2019; Townsend and Keller, 1996; Tison et al., 2010; Matrai et al., 1993).

The Anderson et al. (2001) parameterization assumes chl-a is central to oceanic DMS formation. Previous correlations between chl-a and oceanic DMS, given by the coefficient of determination ($R^2$), range globally from 0.11 to 0.93, with higher latitudes having increased $R^2$ values due to factors like nutrient availability and prolonged summer daylight, coupled with heightened wind speeds (Uhlig et al., 2019; Townsend and Keller, 1996; Tison et al., 2010; Matrai et al., 1993).

L266: The fact that the correlation in the Southern Ocean is high (0.75) does not necessarily support the previous statement. It needs to show the latitudinal dependence (whether the correlations increase with latitude).

*We have included more discussion on this topic within this section*

The Anderson et al. (2001) parameterization assumes chl-a is central to oceanic DMS formation. Previous correlations between chl-a and oceanic DMS, given by the coefficient of determination ($R^2$), range globally from 0.11 to 0.93, with higher latitudes having increased $R^2$ values due to factors like nutrient availability and prolonged summer daylight, coupled with heightened wind speeds (Uhlig et al., 2019; Townsend and Keller, 1996; Tison et al., 2010; Matrai et al., 1993). Gros et al. (2023) estimated an $R^2$ of 0.93 towards sea ice latitudes, while Bell et al. (2021) found chl-a explains just 15% of oceanic DMS variability. Using the Anderson et al. (2001) parameterization in MODIS-DMS, we determined a large $R^2$ of 0.75 in the Southern Ocean. While associating chl-a with oceanic DMS has discrepancies (Gros et al., 2023; Bell et al., 2021), we show that using Anderson et al. (2001) with satellite chl-a data better represents Southern Ocean summertime DMS compared with the MEDUSA configuration.

L270: Half sounds many, but in fact only two (out of four), I think? It might be good to also discuss the lack of DMS models in current ESMs. Also, this paragraph sounds more like discussion than results.

*As we have renamed this section to results and discussion, we have kept this discussion in this section. However, we have added discussion points on the models that contain an interactive oceanic DMS scheme and other oceanic DMS schemes.*

Chl-a is used to calculate oceanic DMS in two of the four ESMs with interactive biogeochemistry in CMIP6 (Bock et al., 2021). These models reveal discrepancies between each other and observed oceanic DMS data sets, indicating ongoing uncertainties in CMIP6 ESMs concerning oceanic DMS and its flux to the atmosphere (Bock et al., 2021). Bock et al. (2021) emphasizes the need for enhanced understanding and observations to

accurately capture DMS–climate feedbacks. CNRM-ESM2-1 adopts an approach considering zooplankton and DMSP rather than chl-a, but its validation is challenging due to limited observational data (Belviso et al., 2012). NorESM2 uses an alternative mechanism for DMS production, by using detritus export production and sea surface temperature (Tjiputra et al., 2020). An oceanic DMS algorithm developed by Galí et al. (2018) includes sea-surface temperature, chl-a, photosynthetically active radiation, and the mixed layer depth, but oceanic DMS has a general overestimation along coastal regions (Galí et al., 2019; Hayashida et al., 2020). Galí et al. (2018) also produced a time series of oceanic DMS over parts of the Northern Hemisphere, finding high inter-annual variability by using chl-a satellite data. Adopting temporally variable oceanic DMS inputs within the model may better reflect inter-annual Southern Ocean variability due to ENSO events and biologically productive years. One such way to achieve this for future projections would be through a stochastic approach of capturing all chl-a years from the satellite (e.g. SeaWiFS and MODIS-aqua) archive.

L273: Suggest to replace "realistic" by "readily available".

done

L285: Is Figure 3 adequate to make the claim that MODIS-DMS is adequate?

*Further discussion of Figures 3 and 4 have been added with more comparisons between the oceanic DMS datasets, with more evaluation between observations and simulations.*

*For TAN1802, we show that Lana reproduces the variability observed by TAN1802 the best but with a high difference in the mean, whereas MODIS-DMS presents similarity to the mean and variability.*

*Additionally, we show that MODIS-DMS best compares to the variability measured by SOAP, whereas the other oceanic DMS datasets used in the model lack this variability. In conjunction with this, MODIS-DMS is also much closer to the mean of SOAP.*

*As a result of these newly added results to the manuscript, we re-phrase this line:*

Having established that oceanic DMS from the MODIS-DMS simulation aligns well with summertime observational voyages as seen in Figure 3, 4, we now assess the sensitivity of atmospheric DMS to various sea-to-air transfer functions (Figure 5, A4).

Fig 5: Suggest to utilize color for distinguishing different DMS sources

*We have used shades of purple, blue, green, and orange to distinguish between MEDUSA, MODIS-DMS, Lana, and Hulswar, respectively.*

L301-302: The two sentences say the same thing. Suggest to remove one. Also, this finding (the choice of DMS source is more important than the choice of flux parameterization) is contradictory to what is written in the abstract (L11).

*Thanks for bringing this to our attention, this has been resolved. We do indeed find that the choice of oceanic DMS datasets produces twice as much variance in atmospheric DMS than the different DMS emissions.*

L303: It is unclear what this sentence is referring to.

*We have clarified to these sentences*

The LM86 flux parameterization was tested with all oceanic DMS sources, as it is currently the parameterization used by default in UKESM1-AMIP. Simulations using LM86 have a spread in average summertime Southern Ocean DMS emissions of 112% (3.3 to 6.9 TgS $Yr^{-1}$). In contrast, simulations using the same oceanic DMS source (MODIS-DMS and Lana) but flux parameterizations (LM86, B17, and W14) have a spread in average summertime Southern Ocean DMS emissions of 51% (MODIS-DMS simulations) to 62% (Lana simulations). The choice of the oceanic DMS source therefore impacts DMS emissions more than the transfer velocity parameterization within these simulations..

L313: Why not compare the model results averaged over the same region as Webb et al. (2019) instead of comparing with the Southern Ocean averages for apple-to-apple comparison?

*We have now compared our flux results with the region from Webb et al. (2019), which forms part of the Western Antarctic Peninsula (WAP). We have also expanded on the discussion from this with comparisons of more studies. This is found in section 3.2 DMS Flux.*

L328: "is likely positively skewed" can be checked to see if the model agrees with this finding by looking at the spatial distribution.

*We have removed the section focussed on the nucleation of aerosol from DMS emissions.*

L332: This paragraph seems more appropriate to be a part of Discussion and conclusions, instead of Results.

*We have renamed the results section: Results and Discussion*

Sec 3.3: How does the distribution of atmospheric DMS differ from that of oceanic DMS emission?

*For the sake of maintaining the clarity of this work, we avoid including this analysis for further unnecessary complications. However below is our response to this question based on our analysis.*

*From all our simulations we calculate a strong correlation between the spatial distribution of the Southern Ocean DMS source (Figure 2a-d) to emissions ($R^2$ of 0.51), indicating a strong oceanic signal in the emissions. The spatial distribution from the oceanic DMS dataset has a stronger signal in the DMS emissions when using a linear flux. MODIS$_{B17}$ has the strongest*

*$R^2$ at 0.85, whereas the signal from Hulswar$_{LM86}$ and Lana$_{W14}$ have the weakest signal, at around 0.26. Implementing a quadratic flux into the DMS source reduces the oceanic DMS signal by around 6% in MODIS$_{B17}$ and up to 24% in Lana$_{B17}$. Oceanic DMS has a slightly weaker relationship with atmospheric DMS (Figure A5), with an average $R^2$ of 0.39, as around 23% of the oceanic DMS signal has been lost through chemical processes and advective transportation. The $R^2$ between the linear fluxes and the DMS source is 0.54, while the linear fluxes of atmospheric DMS concentrations also retain the signal with an $R^2$ of 0.4. This aligns with Zhao et al. (2022), showing how DMS flux is mostly driven by oceanic concentrations.*

*Spatial variability in oceanic DMS drives the distribution of atmospheric DMS, which is much stronger for linear relationships. Constructing oceanic DMS spatial variability proves to be extremely important when assessing the spatial variability of atmospheric DMS. This highlights the need for careful consideration of not only temporal variability but also the need for accurate patterns of spatial variability.*

[Figure]

*Figure 2. summertime (DJF) Oceanic DMS in the Southern Ocean (40 - 60 ◦S). The spatial distribution (a-d) shows the (a) UKESM1 climatology from MEDUSA, (b) the climatology from MODIS-DMS, and observational-based climatologies of (c) Lana and (d) Hulswar. (e) The box plot shows the distribution of each oceanic DMS dataset used.*

[Figure]

*Figure A5. Summertime (DJF) atmospheric DMS concentrations over the Southern Ocean. The spatial distribution shows the simulations with different oceanic DMS concentrations and different DMS flux parameterizations.*

Figure 6: Is this model result representative of the Southern Ocean or the three observational stations?

*This figure has been updated, along with some of the results discussing it, to compare observations more accurately with the simulations. We have taken 5 observational datasets measuring atmospheric DMS over different segments of the Southern Ocean and included them in the updated Figure 6. For observational stations, we have only used measurements taken during the austral summer period. Where the two observational stations are compared with the model, we have calculated the nearest grid cell to the observational station and calculated a temporal mean and standard deviation. In addition to this, we have included three Southern Ocean voyages in our comparison. Hourly model data are compared to voyage data at the appropriate latitude and longitude.*

*Much of the wording has been changed in 3.3 Atmospheric DMS to better reflect these changes. See the updated figure below (Figure 6).*

[Figure]

*Figure 6. Five observational datasets measuring atmospheric DMS concentrations (ppt) are directly compared with the eight simulations (a – e) at the same spatial and temporal resolution. In (a) SOAP and (b) ANDREXII, we follow both voyages using the nearest grid cell along each hour of the simulations, matching the timescales in 2012 and 2019. For comparing the simulations with the (c) SOIREE voyage, we also follow this voyage in an hourly timescale, but due to the voyage being outside our study period, we average this over all 10 years. The two observational stations used are (d) Cape Grim and (e) King Sejong Station. We calculate the nearest grid-cell for each simulation to the observational station and constructed an average over 10 years along with a temporal standard deviation. From this, we construct an overall average (f) and standard deviation for all observational measurements and simulations which can be compared directly to these observations.*

L365 and Fig 7: Suggest to delete the results of PMOA. As the authors stated (L378), the results of PMOA require further investigation that does not seem to fit into the scope of this study. Adding this result would require additional info on intro, methods, and discussion.

*We have removed this section.*

L374: Suggest to rewrite this sentence. It states about CMIP6 but the paper cited is from 2011, many years before CMIP6.

*This sentence has been removed.*

L385: Suggest to cite a paper that compares nudged vs non-nudged runs to support this argument.

*We have removed this sentence and moved it to the methods section, including a citation to Pithan et al. (2022).*

L402: How is this different from L399?

*This sentence has been removed.*

L420: Why recommend Lana while there is an updated climatology?

*This sentence has been updated, removing suggestions of a particular oceanic DMS dataset, but rather focusing on the transfer velocity used:*

In future, we recommend that models use up-to-date DMS-specific relationships such as B17.

Appendix A: Figures should be labeled as A1-4 instead of A-D1

*Fixed.*

**REVIEWER 2:**

- That using a satellite monthly chlorophyl to drive seawater DMS concentration is superior to using a climatological seawater DMS field seems not unreasonable in principle. However the validations using only two cruises are not very convincing to me. Can the authors include more validation datasets? For example the SO-GasEx dataset is within the 10-year window and includes seawater/atmospheric DMS as well as DMS flux.

*Unfortunately, the SO-GasEx voyage (29 February to 12 April 2008) falls outside our study's 10-year simulation window (2009 to 2018). Incorporating it would involve re-running the simulations to start earlier.*

*However, we have lengthened the simulations to overlap with the ANDREXII voyage, from February to April 2019 (Wohl et al., 2020) solely focusing on atmospheric DMS concentrations.*

- There have been more recent seawater DMS parametrizations than Anderson et al. 2001. The authors should include them or explain why they aren't considering these more recent developments. They should also acknowledge that seawater DMS doesn't just depend on Chla, but is also sensitive to a number of other biological parameters.

*While we acknowledge that Anderson et al. (2001) is not the most recent approach to modeling oceanic DMS concentrations, it remains a seminal work in the field and serves as the foundational DMS parameterization within the UKESM1 (e.g. Bock et al., 2021; Sellar et al., 2019; Simo and Dachs., 2002; Vallina and Simo., 2007). Our study focuses on the impact of atmospheric DMS variability rather than the intricacies of oceanic DMS formation. Therefore, we opted for the UKESM1's native parameterization by Anderson et al. (2001) for consistency to align with the model's existing framework, and to better compare using a satellite-derived chlorophyll-a dataset compared with MEDUSA.*

*We have added justification of our approach as described:*

**2.2 Oceanic DMS (Methods):**

In UKESM1, oceanic DMS concentrations are calculated using a diagnostic method from Anderson et al. (2001), using surface daily shortwave radiation (J), dissolved inorganic nitrogen (Q), and chl-a (C):

$$Oceanic\ DMS\ =\ a, for\ log(CJQ)\ \leq\ s$$

( 2 )

$$Oceanic\ DM\ S\ =\ b[log(CJQ)\ -\ s]\ +\ 1, f\ or\ log(CJQ)\ >\ s$$

(2)

The parameter values are a=1, b=8, and s=1.56, as described by Sellar et al. (2019). Q, chl-a, and J are averaged from CMIP6 for the MEDUSA climatology. The Anderson et al. (2001) parameterization produces positive biases in DMS over the Southern Ocean within MEDUSA (Bock et al., 2021) due to the set minimum oceanic concentration of 1, which leads to large average DMS concentrations (Yool et al., 2021; Bock et al., 2021). Recent research suggests that chl-a may not be an appropriate proxy for oceanic DMS (Uhlig et al., 2019; Bell et al., 2021), and future work will explore alternative methods for calculating oceanic DMS within UKESM1. Nonetheless, chl-a is widely used by CMIP6-era models to calculate oceanic DMS, and we explore here whether using an observationally derived chl-a concentration field leads to changes in the spatial and temporal variability of atmospheric DMS. Monthly-mean chl-a concentrations from the Moderate Resolution Imaging Spectroradiometer (MODIS)-aqua satellite instrument were used to construct a time series of oceanic DMS between 2009–2018 (Table 1; Hu et al., 2019; O'Reilly and Werdell, 2019). This time series, which we term the 'MODIS-DMS' data set, is calculated offline using the same diagnostic parameterization as Equations 1 and 2. The J and Q used to calculate MODIS-DMS remain the same as MEDUSA. Through this, we capture spatial and interannual chl-a variability, indicating biological productivity. Bi-linear interpolation

is used to fill in small gaps (around 1% for monthly averages) of spatial chl-a data. Oceanic DMS concentrations are masked where they coincide within the sea-ice zone from HadISST.

*In addition to this, we also further justified the use of using chl-a to calculate oceanic DMS in our study, as outlined below:*

The Anderson et al. (2001) parameterization assumes chl-a is central to oceanic DMS formation. Previous correlations between chl-a and oceanic DMS, given by the coefficient of determination ($R^2$), range globally from 0.11 to 0.93, with higher latitudes having increased $R^2$ values due to factors like nutrient availability and prolonged summer daylight, coupled with heightened wind speeds (Uhlig et al., 2019; Townsend and Keller, 1996; Tison et al., 2010; Matrai et al., 1993). Gros et al. (2023) estimated an $R^2$ of 0.93 towards sea ice latitudes, while Bell et al. (2021) found chl-a explains just 15% of oceanic DMS variability. Using the Anderson et al. (2001) parameterization in MODIS-DMS, we determined a large $R^2$ of 0.75 in the Southern Ocean. While associating chl-a with oceanic DMS has discrepancies (Gros et al., 2023; Bell et al., 2021), we show that using Anderson et al. (2001) with satellite chl-a data better represents Southern Ocean summertime DMS compared with the MEDUSA configuration.

Comparisons in atmospheric DMS neglects the time component entirely. Maybe the authors can further expand their 10-year window to facilitate comparisons with those earlier and very recent atmospheric DMS measurements?

*We believe this updated section, along with the following section on 3.4 Effects from Inter-annual and Spatial Variability effectively resolves this weakness. As stated above, we have also expanded our simulation window to include the ANDREXII voyage in 2019, as per the recommendation to allow more like-to-like comparisons. Figure 6 has been updated, along with some of the results discussing it, to more accurately compare observations with the simulations. We have taken 5 observational datasets measuring atmospheric DMS over different segments of the Southern Ocean. For observational stations, we have only used measurements taken during the austral summer period. To compare these observations more appropriately than previously done, to our simulations, we have considered the temporal and spatial dimensions of these measurements. Where the two observational stations are compared, we have calculated the nearest grid cell to the observational station and constructed an average and temporal standard deviation. In addition to this, we have included three Southern Ocean voyages in our comparison. By tracking the latitude and longitude of each voyage in the hourly timescale in our simulations, we calculate an average for each simulation across each voyage. We also calculate the spread of each simulated voyage through the standard deviation, to compare fairly with the voyage dataset. From this, we calculate an overall average and standard deviation across each simulation and observational dataset, allowing for a more like-to-like comparison of simulations against observations.*

*Much of the wording has been changed in Section 3.3 Atmospheric DMS to better reflect these changes. See the updated figure below (Figure 6). We have also included individual observation comparisons with each simulation in Figure A5.*

[Figure]

Figure 6. Five observational datasets measuring atmospheric DMS concentrations (ppt) are directly compared with the eight simulations (a – e) at the same spatial and temporal resolution. In (a) SOAP and (b) ANDREXII, we follow both voyages using the nearest grid cell along each hour of the simulations, matching the timescales in 2012 and 2019. For comparing the simulations with the (c) SOIREE voyage, we also follow this voyage in an hourly timescale, but due to the voyage being outside our study period, we average this over all 10 years. The two observational stations used are (d) Cape Grim and (e) King Sejong Station. We calculate the nearest grid-cell for each simulation to the observational station and constructed an average over 10 years along with a temporal standard deviation. From this, we construct an overall average (f) and standard deviation for all observational measurements and simulations which can be compared directly to these observations.

- What are the impacts of these different DMS emission simulations on aerosols, CCN, AOD, etc? Can authors validate those?

*We have added in a new subsection at the end of the paper, called Aerosol and cloud response. This is to quantify the effects of the changes made from this manuscript to the aerosol and cloud properties. Although we must emphasise that much of the processes involved in aerosol formation and cloud processes are beyond the scope of this study, we hope to provide useful insight into how these changes affect clouds and aerosols. We also show comparisons of observational datasets during DJF to further validate our study. The new section reads as follows:*

**3.5 Aerosol and cloud response**

Figure 9 shows the effect on cloud and aerosol properties of changing the atmospheric DMS distribution. Changing the atmospheric DMS concentration yields little change to CCN, CDNC or AOD. This suggests that these variables are significantly influenced by factors such as sea spray aerosol and the atmospheric oxidation pathways that convert DMS to sulfate aerosol (Revell et al., 2021; Fossum et al., 2020). Changes to the DMS source increase the spread in simulated CCN and CDNC over the Southern Ocean rather than changing the mean DMS emissions, which is consistent with our findings for atmospheric DMS concentrations. Altering the DMS source affects AOD by 73% more than DMS emissions over the Southern Ocean, emphasizing the role of the ocean in producing atmospheric DMS. Box plots of AOD, CCN, and CDNC (Figure 9e, a, c) show that the simulations do not capture the maxima in CDNC, CCN or AOD over the Southern Ocean.

[Figure]

*Figure 9. Summertime climatology between 65◦ S to 40◦ S showing the (a,b) cloud droplet number concentrations, (c,d) cloud conversation nuclei (800 m in altitude), and (d,e) aerosol optical depth at 550 nm. The violin plots (a,c,e) represent all spatial and temporal data points across the 10 years over the Southern Ocean in DJF. The lowest 1% of values are excluded from the violin plots. In (b,d,f) the grey lines represent observational datasets where (b) Grosvenor et al. (2018) (dashed) and Bennartz and Rausch (2017) (solid) are shown for CDNC, (d) Choudhury and Tesche (2023) is shown at 818m, and (f) AOD climatology by the MODIS satellite-retrieval is shown (Platnick et al., 2017). The error bars represent one standard deviation either side of the observational mean.*

**Specific comments:**

Please make careful distinction between flux and Kw. The authors have incorrectly interchanged their usage a number of times in the paper.

*Thank you for letting us know of the incorrect terms being used throughout this work. We have corrected where flux is stated but transfer velocity is meant.*

Overall the writing isn't very concise. I suggest further proof-reading from all authors.

*We have worked on making this manuscript more concise.*

Line 11-12. Why, given that the spread is greater among different oceanic DMS datasets?

*Thank you for bringing this to our attention, we accidentally reversed the order. This has now been corrected:*

The choice of oceanic DMS source has a larger influence on atmospheric DMS than the choice of DMS emission.

Line 12-15. Be more exact. It's a linear relationship between gas transfer velocity and wind speed. More so than between flux (emission) and wind, since flux also depends on seawater DMS concentration

*We have applied these changes to these sentences to provide greater clarity:*

Simulations testing different sea-to-air transfer velocity parameterizations show that simulating a linear dependence of DMS gas transfer velocity as a function of wind speed results in a more accurate representation of atmospheric DMS distributions than using quadratic relationships.

Line 25-26. DMS isn't just produced by phytoplankton. There's also bacterial production. So suggest replacing 'phytoplankton' with 'marine biota' and update references accordingly.

*We have made these amendments throughout, with the updated references shown below.*

(Keller et al., 1989; Bates et al., 1987; Kiene and Bates, 1990; Curson et al., 2011)

Line 30. This paragraph talks about how ESMs represent DMS emission. But I think it'd be better to first talk about how DMS emission is estimated in general. i.e. flux = Kw * deltaC ~= Kw * [seawaterDMS]. Then you can talk about the different parametrizations of Kw (the gas transfer velocity), as well as the different ways [seawaterDMS] is estimated.

*We have included a section discussing different DMS emissions, which were moved from the methods section. We have also made new additions to this paragraph, including the reference to the flux from the gas transfer velocity.*

*The introduction paragraph has been changed from:*

DMS is emitted from the ocean to the atmosphere and has a strong dependence on the surface wind speed (e.g. Fairall et al., 2011). A wealth of research has focused on better understanding the relationship between atmospheric DMS and wind speed (Vlahos and Monahan, 2009; Zavarsky et al., 2018; Blomquist et al., 2017; Wanninkhof, 1992, 2014; Nightingale et al., 2000; Liss and Merlivat, 1986; Goddijn-Murphy et al., 2016; Ho et al., 2006; Bell et al., 2015). However, the uncertainty in this relationship remains high particularly within the Southern Ocean due to a lack of observational data (e.g. Elliott, 2009; Smith et al., 2018; Zhang et al., 2020), particularly for wind speeds ≥ 13 ms−1 (Blomquist et al., 2017). Recently, significant progress has been made as recent literature has established that DMS flux has a linear relationship with wind (Goddijn-Murphy et al., 2016; Blomquist et al., 2017; Bell et al., 2015), while Earth System Models continue to use older quadratic relationships to represent DMS emissions Bock et al. (2021).

*to*

The flux of DMS from the ocean to the atmosphere depends on the gas transfer velocity, which in turn depends on the surface wind speed (e.g. Fairall et al., 2011). Many DMS flux parameterizations have been developed, but most use transfer velocities measured for gases other than DMS (Wanninkhof, 1992, 2014; Nightingale et al., 2000; Liss and Merlivat, 1986). Some studies, including Blomquist et al. (2017) and Yang et al. (2011), used DMS measurements to derive a relationship between wind speed and DMS. Depending on the solubility of the gas measured, gas transfer velocities typically have a linear or quadratic dependence on wind speed. Linear relationships best represent gases with intermediate solubilities, such as DMS (e.g. Blomquist et al., 2017; Goddijn-Murphy et al., 2016; Bell et al., 2015; Yang et al., 2011; Huebert et al., 2010), while quadratic equations are better suited for highly soluble gases like $CO_2$ (Wanninkhof, 2014; Nightingale et al., 2000; Wanninkhof, 1992).

Uncertainty in DMS emissions remains high, particularly in the Southern Ocean region where wind speeds are high and observational data sparse (e.g. Elliott, 2009; Smith et al., 2018; Zhang et al., 2020). Earth System Models use a variety of transfer velocities to represent DMS emissions (Bock et al., 2021). UKESM1 uses the Liss and Merlivat (1986) parameterization even though it was constructed for gases other than DMS.

Line 38-39. Not sure what 'simulate biases' means here

*Reworded sentence from:*

CMIP6 models simulate biases in oceanic DMS production compared with observational climatologies of DMS in the Southern Ocean region (Bock et al., 2021).

*To:*

Bock et al. (2021) evaluated oceanic DMS in CMIP6 models and found that all models are biased in comparison with observational climatologies of DMS in the Southern Ocean region.

Line 47. Suggest adding 'to some degree' after 'events', given the complexity in biological response.

 *done*

Line 49. Not between atmospheric DMS and wind speed, but between the gas transfer velocity and wind speed.
Some of the references aren't the most appropriate:
Vlahos and Monahan 2009 interpreted other groups' measurements
 Wanninkhof 1992, 2014 focused on CO2, not DMS. Nightingale et al. 2000 and Ho et al. 2006 focused on 3He/SF6, not DMS. Liss & Merlivat 1986 synthesized a range of lab/field works, but not specifically of DMS. Those parametrizations are often used in ESMs for estimating DMS flux, inappropriately. So it's useful to introduce them here, but with a clarification.
 Also, suggest adding some of the original literature on DMS gas exchange:
doi:10.1029/2004GL021567
doi:10.1029/2009GL041203
doi:10.1029/2010JC006526

*We have made substantial modifications to this paragraph, as shown above. This is shown below:*

The flux of DMS from the ocean to the atmosphere depends on the gas transfer velocity, which in turn depends on the surface wind speed (e.g. Fairall et al., 2011). Many DMS flux parameterizations have been developed, but most use transfer velocities measured for gases other than DMS (Wanninkhof, 1992, 2014; Nightingale et al., 2000; Liss and Merlivat, 1986). Some studies, including Blomquist et al. (2017) and Yang et al. (2011), used DMS measurements to derive a relationship between wind speed and DMS. Depending on the solubility of the gas measured, gas transfer velocities typically have a linear or quadratic dependence on wind speed. Linear relationships best represent gases with intermediate solubilities, such as DMS (e.g. Blomquist et al., 2017; Goddijn-Murphy et al., 2016; Bell et al., 2015; Yang et al., 2011; Huebert et al., 2010), while quadratic equations are better suited for highly soluble gases like CO2 (Wanninkhof, 2014; Nightingale et al., 2000; Wanninkhof, 1992).

Uncertainty in DMS emissions remains high, particularly in the Southern Ocean region where wind speeds are high and observational data sparse (e.g. Elliott, 2009; Smith et al.,

2018; Zhang et al., 2020). Earth System Models use a variety of transfer velocities to represent DMS emissions (Bock et al., 2021). UKESM1 uses the Liss and Merlivat (1986) parameterization even though it was constructed for gases other than DMS.

line 54. Again, the linear relationship is between the gas transfer velocity K and wind speed, not between flux and wind speed. Also, please add doi:10.1029/2009GL041203 doi:10.1029/2010JC006526 here

*We have modified this throughout the manuscript to depict this relationship accurately. We have also added these references to this section.*

line 63. There are a few other schemes in addition to Anderson et al. 2001 that makes use of Chla. Why only Anderson et al. 2001? Why not adding e.g. Gali et al. 2018?

*Please see our response above justifying the use of Anderson et al. (2001).*

Line 85. That simulation is 10 years long has been said a couple of times now, but without explanation. Presumably it's to coincide with DMS observations.

*We have added text explaining why this time period was chosen. The simulations are 10 years in length to provide sufficient time to capture interannual variability.*

This period was chosen to coincide with the availability of recent DMS observations (Section 2.4).

Line 106. So if I'm understanding correctly, this is a climatology because Chla and nutrients in UKESM1 are climatological?

*We have removed this sentence as it was confusing and added clarity to the following sentence as such:*

In UKESM1, oceanic DMS concentrations are calculated using a diagnostic method from Anderson et al. (2001), using surface daily shortwave radiation (J), dissolved inorganic nitrogen (Q), and chl-a (C):

$$Oceanic\ DMS\ =\ a, for\ log(CJQ)\ \leq\ s$$

( 3 )

$$Oceanic\ DM\ S\ =\ b[log(CJQ)\ -\ s]\ +\ 1, for\ log(CJQ)\ >\ s$$

(2)

The parameter values are a=1, b=8, and s=1.56, as described by Sellar et al. (2019). Q, chl-a, and J are averaged from CMIP6 for the MEDUSA climatology.

Line 119-123. This motivation should've been laid out already in intro, and doesn't warrant repeating here.

*These two sentences have been removed and included in the introduction.*

Line 126. Can you add/cite an accuracy number here? i.e. x%?

*More details on the past validation of chl-a in the Southern Ocean have been included:*

 Several studies have validated the MODIS-aqua Ocean Color chl-a retrieval, finding it to generally underestimate Southern Ocean conditions (Zeng et al., 2016; Haëntjens et al., 2017; Jena, 2017). Satellites can also overestimate chl-a measurements due to the scattering of light from aerosols (Schollaert et al., 2003). However, Marrari et al. (2006) found satellite chl-a is accurate within the Southern Ocean during summer. Therefore the high spatial and temporal availability of summertime data makes chl-a a viable option for estimating phytoplankton productivity and oceanic DMS.

*To*

In general, the MODIS-aqua Ocean Color chl-a retrieval underestimates Southern Ocean chlorophyll concentrations (Zeng et al., 2016; Haëntjens et al., 2017; Jena, 2017; Gregg and Casey, 2007; Johnson et al., 2013). Simulated oceanic DMS may therefore be systematically underestimated. Nonetheless, the high spatial and temporal availability of chl-a observations during summertime makes it useful to explore spatiotemporal variability in atmospheric DMS.

Line 132. A lot of this paragraph could've been laid out in intro.

*Much of the introduction has been changed to include much of the transfer velocity background (see next comment).*

Line 141. some of this has been talked about already in intro. I also find the rest of this paragraph wordy. Table 2 summarizes everything well. The main message here is just to offer a range of emissions.

*Significant changes have been made to this paragraph. Much of it has been moved to the introduction section, and re-written. The rest has been reduced with improved clarity. The changes made are from:*

[revised manuscript text omitted]

Figure 1 y-axis label is incorrect. It's gas transfer velocity, not flux

*Fixed.*

Line 209. Does this imply the mean Chla in MEDUSA is very different from the mean Chla from MODIS? Ok this is explained in the following paragraph..

Figure 3. grey is derived from observations. Colors are the model simulations?

*We have now clarified this in the figure caption.*

Line 262. Chla from MODIS is monthly, while the comparison with in situ data is I think on hourly timescales. In reality Chla is not constant over a month. How does this mismatch in timescale affect the comparison?

*The potential mismatch in timescales between monthly MODIS Chl-a data and hourly in situ data may introduce variability in the comparisons. However, our focus was on capturing broader trends and patterns rather than short-term fluctuations. By using the MODIS-DMS dataset, we aimed to accurately simulate the spatial and temporal distributions of oceanic DMS concentrations comparable to observations.*

Comparison in atmospheric DMS: measurements were taken from near the sea surface (e.g. 20 m height). What height is the model atmospheric DMS taken from?

*We add in this sentence in the model description:*

Atmospheric DMS concentrations are analyzed at the lowest model level, at 20 m during DJF, which is the most productive season for DMS (Deppeler and Davidson, 2017; Jarníková and Tortell, 2016).

Line 284. Looking at Figs. 3 and 4, I'm not sure if 'good' is the right word here. Perhaps better to say something like 'agree within x% in the mean'

*Further discussion of Figure 3 and 4 are done with more comparisons between the oceanic DMS datasets, with definite assessments. For TAN1802, we show how Lana reproduces the variability observed by TAN1802 the closest, whereas MODIS-DMS presents similarity to the mean and variability.*

*Additionally, we show how MODIS-DMS best produces the variability from SOAP, whereas the other datasets lack this variability. In conjunction with this, MODIS-DMS also is much closer to the mean of SOAP.*

*As a result of these newly added results to the manuscript, we re-phrase this line:*

Having established that oceanic DMS from the MODIS-DMS simulation aligns well with summertime observational voyages as seen in Figure 3, 4, we now assess the sensitivity of atmospheric DMS to various sea-to-air transfer functions (Figure 5, A4).

Line 299, 302. Again, Kw (not flux) parametrization

*These have been reworded*

Line 303, 304. Sentence unclear

*We have clarified this sentence by some additions:*

The LM86 flux parameterisation was tested with all oceanic DMS sources, as it is currently the parameterisation used by default in UKESM1-AMIP. Simulations using LM86 have a spread in average summertime Southern Ocean DMS emissions of 112% (3.3 to 6.9 TgS Yr−1). In contrast, simulations using the same oceanic DMS source (MODIS-DMS and Lana) but flux parameterizations (LM86, B17, and W14) have a spread in average summertime Southern Ocean DMS emissions of 51% (MODIS-DMS simulations) to 62% (Lana simulations). The choice of the oceanic DMS source therefore impacts DMS emissions more than the transfer velocity parameterization within these simulations.

Figure 5. why does MEDUSALM86 have quite a large stdev in flux, when the simulated seawater DMS from MEDUSA has the smallest stdev?

*This is an interesting point, as Figure 2 from the manuscript shows the box plot distribution of MEDUSA as being relatively small. When reproducing this for the flux (Figure A4, also shown below), we also see that MEDUSA produces the smallest spread of data. Although the standard deviation is high, the actual spread of data is the lowest across the simulations. This is attributed to the high mean value from MEDUSA, which results in the +-1 standard deviation naturally being relatively high. Although the CoV from MEDUSA is lower than the other simulations, meaning that the spatial and temporal variability from MEDUSA is lower. We have now included this figure in the appendix and referenced it in the text:*

we now assess the sensitivity of atmospheric DMS to various sea-to-air transfer functions (Figure 5, A4).

[Figure]

*Figure A4. DMS emissions for all simulations over the 10 years using daily data in the summertime (DJF) over the Southern Ocean as a box plot.*

Line 324-325. Is this statement about nucleation up to date? Does it consider tertiary nucleation involving amines/ammonia, for example? Also, the background aerosol surface area and thus condensation sink must be key here.

*We agree that the nucleation of particles involves multiple complicating factors, especially in an area such as the Western Antarctic Peninsula. While our focus here is on DMS-driven nucleation, we acknowledge that other compounds, such as amines and ammonia, can also play significant roles in tertiary nucleation processes.*

*However, to reduce the size of the manuscript and make it more concise, we have removed this section about nucleating particles from DMS.*

Line 335-337. I'm not sure that this is due to the linear relationship between Kw and wind speed. Because the monthly Chla field doesn't capture the true variability in Chla, it seems likely that some peak biological events will be missed. Furthermore Chla is far from a perfect descriptor of seawater DMS, as different species of marine biota, bloom senescence, grazing, etc can all affect seawater DMS concentrations.

*We agree with this statement, in that Chl-a is not a perfect descriptor of oceanic DMS, but the current spatial and temporal observational data that tracks ocean DMS best, such as marine biota, bloom senescence, grazing, etc are lacking. The use of chl-a is currently the best available proxy (e.g. Gali et al., 2018). An additional figure (Figure 7) has been added to the manuscript to further evaluate the importance of annually varying oceanic DMS concentrations against climatologies within our MODIS-DMS dataset, where only the chl-a is time-varying. Figure 7c illustrates few differences between atmospheric DMS concentrations for each year, but is overwhelmingly consistent between the two simulations.*

*Although chl-a would not capture the full diurnal variability due to current satellite constraints, we disagree that the current chl-a used in this study does not capture the true spatial and temporal variability, shown in Figure 8.*

*The statement below more accurately outlines our findings*

We demonstrate that linear DMS transfer velocities represent the DMS flux ranges better than the quadratic W14 flux when compared to Southern Ocean observations.

Figure 6. The authors have been arguing about the benefits in including temporal variability via using the MODIS Chla. Yet the comparison here is vs. mean observed atmospheric DMS, much of which lies outside of the simulation window. Why not also show the equivalent of Figures 3, 4, B1, C1, but for atmospheric DMS?

*Please see the comment made previously about Figure 6, where we have responded in full, including additional figures.*

Figure 7. I thought MEDUSA has higher Chla than MODIS-DMS, so how come the latter produces higher primary organics? In general I find the discussion about primary organics off-topic. I suggest focusing the discussion on how using MODIS Chla may improve the representation of DMS emission over climatology, and what the remaining shortcomings/uncertainties are in this approach. i.e. how well can DMS be simulated when Chla is known, and at what temporal/spatial scales? What about other factors including phytoplankton speciation, bacterial production, zooplankton grazing, viral lysis?

See also doi: 10.3389/fmars.2020.596763

*We have removed this figure and the section on PMOA.*

Also, can the authors validate their simulations against other parameters such as aerosol non-seasalt sulfate, aerosol number concentration, CCN, AOD?

*We have added a new section based on validating our simulations with AOD, CCN, and CDNC observations, shown below. We have also added in new information in the methodology to coincide with the extra variables which have been analyzed.*

**3.5 Aerosol and cloud response**

Figure 9 shows the effect on cloud and aerosol properties of changing the atmospheric DMS distribution. Changing the atmospheric DMS concentration yields little change to CCN, CDNC or AOD. This suggests that these variables are significantly influenced by factors such as sea spray aerosol and the atmospheric oxidation pathways that convert DMS to sulfate aerosol (Revell et al., 2021; Fossum et al., 2020). Changes to the DMS source increase the spread in simulated CCN and CDNC over the Southern Ocean rather than changing the mean DMS emissions, which is consistent with our findings for atmospheric DMS concentrations. Altering the DMS source affects AOD by 73% more than DMS emissions over the Southern Ocean, emphasizing the role of the

ocean in producing atmospheric DMS. Box plots of AOD, CCN, and CDNC (Figure 9e, a, c) show that the simulations do not capture the maxima in CDNC, CCN or AOD over the Southern Ocean.

[Figure]

*Figure 9. Summertime climatology between 65◦ S to 40◦ S showing the (a,b) cloud droplet number concentrations, (c,d) cloud conversation nuclei (800 m in altitude), and (d,e) aerosol optical depth at 550 nm. The violin plots (a,c,e) represent all spatial and temporal data points across the 10 years over the Southern Ocean in DJF. The lowest 1% of values are excluded from the violin plots. In (b,d,f) the grey lines represent observational datasets where (b) Grosvenor et al. (2018) (dashed) and Bennartz and Rausch (2017) (solid) are shown for CDNC, (d) Choudhury and Tesche (2023) is shown at 818m, and (f) AOD climatology by the MODIS satellite-retrieval is shown (Platnick et al., 2017). The error bars represent one standard deviation either side of the observational mean.*

Line 399-400. This seems correct but is in conflict with abstract

*We have resolved this by changing the abstract, as stated previously. Thanks for bringing this to our attention.*

Line 402. Why is that, given the fact that the flux should be more directly linked to atmospheric DMS than is seawater

*This sentence has been removed.*

Line 407. I agree with this statement, yet the authors subsequently advocate for the use of LM86, which is neither recent nor specially suitable for DMS.

*This sentence has been updated to remove LM86 and more accurately depicts our recommendation based on our results:*

In future, we recommend that models use up-to-date DMS-specific relationships such as B17.

Figure D1. SOExchange isn't the right acronym and data source isn't cited.  It's SO-GasEx (doi:10.1029/2010JC006526).  Also for (j) and (k) here it's not obvious if the authors are showing observations or simulations.

*This figure has been removed to allow more focus on the additional figures in the main text and appendix.*

---

## Author Response (AR2)

We appreciate the reviewers for their constructive comments and suggestions. Below we respond to each comment; reviewer comments are shown in black, our response is in *red italics*, and revised text is in blue. The line numbers we refer to are within the tracked changes document.

**Response to Referee #1**

I appreciate the authors for their thorough responses to my previous review. The manuscript has been improved substantially. I am happy to recommend 'accepted subject to technical corrections'. Below are my suggested minor/editorial corrections for Abstract and Conclusions:

*We thank the reviewer for their comments. The abstract and conclusions sections have been re-written as per the request from Reviewer 2. Therefore, many of these sentences have been re-phrased or deleted. Where this is the case, I will refer to it as 'changed'.*

L1: Replace 'complex and dynamic and driven' by 'complex, dynamic, and driven'

*Done*

L2: Replace 'leads' by 'lead'

*Changed*

L4: Remove 'nudged to observations'

*Done*

**L5: Add a few words to describe what has been 'tested' (the sensitivity)**

*L6 now reads:*

*We tested the sensitivity of atmospheric DMS to four seawater DMS data sets and three DMS transfer velocity parameterizations.*

L12: Replace 'DMS emission' by 'flux parameterization' because emission depends on source (source x flux parameterization)

*L14 - This sentence has been changed to:*

*We find that the choice of oceanic DMS data set has a larger influence on atmospheric DMS than the choice of DMS transfer velocity.*

L17: Either remove the sentence 'As a precursor …' or move it to the beginning of the abstract, as it is a background information and not the finding of this study

*We have removed this sentence.*

L352: Add the year to 'Gali et al'

*done*

L367: Replace 'B17' by 'Blomquist et al. (2017)' for clarity

*done*

**Response to Referee #2**

The revised version of this paper has addressed many of the questions raised by me and the other reviewer and does read better. However, I think the writing can be further improved to tease out the key messages/findings. This is especially important in the title, abstract, and conclusions, but is also relevant to the other sections.

Climatology of seawater DMS concentrations such as Lana et al and Hulswar et al, containing monthly concentration field for each grid cell, have been used historically in many ESMs to drive DMS flux. Doing so requires a choice of the gas transfer velocity (K), which comes with significant uncertainty. The use of satellite Chla (and other ancillary parameters) also allow for a description of the interannual variability in seawater DMS concentration and so flux, which cannot be captured by using a climatology.

The main motivations of this work (according to my reading of the paper, and according to the authors' replier to reviewers) are to see 1) how sensitive are DMS flux and concentrations (in water and in air) to these different options of K and DMS concentration fields, and 2) the impact of interannual variability on atmospheric DMS concentration. These motivations are fine, but the abstract/conclusion do not currently reflect point 2 (to me the key motivation).

Currently the abstract/conclusion really focus on the finding that a linear wind speed dependence in K is superior to a quadratic dependence. While this is a useful message (especially to the earth system modellers), it really isn't a new finding – in situ observations more than a decade ago show that K of DMS and wind speed has a near linear relationship.

The current conclusion also states that by combining satellite Chla with a seawater DMS parameterization (Anderson et al 2001), one captures the spatial/temporal variability in atmospheric/seawater DMS better than using a climatology. To me, it's clear that this approach would provide more interannual variability than a climatology. However, based on the evidence shown it's not obvious that this approach captures the spatial variability 'better' than the climatology. As seen in Fig 2, the seawater DMS concentration field driven by Chla/Anderson is very different from the observation-based climatology. And the limited comparisons with in situ observations do not yield clear cut evidence to me that MODIS-DMS is 'better' or more accurate (but better than MEDUSA? Probably). For example it seems to give a lower mean (in seawater DMS, DMS flux, and atmospheric DMS) than the observation-based climatology. Are the authors arguing that the climatology itself is biased high?

This discrepancy is probably in part due to the decision of using the Anderson parametrization, which is >20 years out of date and does not represent the state-of-the-art. I would've liked to see the authors testing some more recent parametrizations of seawater DMS concentrations. If for whatever reason the authors cannot do so, fine, but then the authors should refrain from statements that suggest MOIDS-DMS captures the mean or spatial variability in seawater DMS better than climatology. They should focus more on the interannual variability component.

I suggest the authors to rewrite the abstract/conclusion with the following components (and revise the paper to reflect/back up these points):

- Sensitivities of DMS concentrations (water/air) and flux to the different parametrizations of K and seawater DMS concentration fields

- Various validations using in situ observations

- Then the key take home messages:

1) That use a linear wind speed dependence in K (based on DMS measurements) is better than using a quadratic dependence. Again, a useful message, but don't labour over it.

2) the seawater DMS concentration field driven by Chla/Anderson is very different from the observation-based climatology. Probably need (to incorporate) more observations to say conclusively which is more accurate

3) interannual variability (a key motivation for their MODIS-DMS approach). The authors seem to suggest currently in the conclusion that it's not very important, but I don't think their estimates capture the entire interannual variability. See detailed comments below.

4) Atmospheric impact: the authors seem to suggest that Nd/CCN/AOD are not very sensitive to the different DMS emissions, but I think Figure 9 shows otherwise. While none of the emission options explains the gaps between simulations and satellite derived Nd/CCN/AOD, among the simulations themselves I think there's quite a range in especially Nd and CCN, which is worth stating.

Finally, they should probably acknowledge that future work should adopt more recent parametrization of seawater DMS concentration.

We thank the reviewer for this comprehensive overview and detailed suggestions on improvements to the conclusion and abstract.  To better reflect our key motivation and results within the conclusion and abstract we have focused more on including inter-annual variability of DMS, and the sensitivity of different oceanic DMS data sets and transfer velocities used to atmospheric DMS concentrations. We have also re-written the abstract to fit the ACP requirements of around 250 words and focussed the abstract.

Changes from these specific comments are located in the tracked changes document for:

Lines 396 to 412 for the conclusion and lines 1 to 29 for the abstract.

Specific comments.

Overall, the writing is still not very precise. Please be specific whether the sentences are talking about DMS concentrations (water/air) or flux. And make clear distinctions between DMS flux (~=K * [DMS]seawater) and DMS transfer velocity (K).

We have focused the paper on the main results, especially in the conclusion.

Title. To many, the word 'source' can be synonymous with 'emission'. Here the authors really mean 'surface seawater concentration'

We have changed the term 'source' in the title with 'oceanic DMS concentrations'.

Line 9. 'Same gas transfer velocity', not 'same sea-to-air flux'

done

Line 10. DMS FLUX varies …

Changed from 'DMS varies' to 'DMS emissions vary'.

Line 11. 'different gas transfer velocity', not 'differing sea-to-air flux'

done

Line 12. The phrase 'DMS source' implies a flux to me. But the authors actually mean a different seawater DMS concentration dataset

This has now been changed to oceanic DMS data set

Line 16. 'recently developed transfer velocity parametrizations' is presumably a linear one? The last two sentences have some repetition

We have added 'linear' to this sentence.

Line 17-18 this could be moved to near the top of the abstract

We have removed this sentence as recommended by Reviewer 1

Line 18-20 I find it strange that one of the main messages in the abstract is that the seawater DMS datasets and transfer velocity parametrizations are poorly constrained. Isn't the point of this paper to improve the representations of these processes?

L21 This sentence has now been changed to better reflect that it is poorly constrained within current climate models:

This work highlights that the seawater DMS data sets and transfer velocity parameterizations for DMS currently used in climate models are poorly constrained for the Southern Ocean region.

Line 29-30; 35-37. These descriptions of marine DMS cycling are pretty vague. I suggest for line 29, remove the 'controlled by marine biota statement', and move that to line ~36. More accurately, phytoplankton produces DMSP, and bacteria+ phytoplankton consume DMSP to produce DMS. Thus one expects some correspondence between chlorophyll a and DMS, but not a perfect one.

L41 We have modified to sentence to read as:

Marine biogenic activity, controlled by marine biota, plays a key role in chlorophyll-a (chl-a) production and is considered to be a key driver of oceanic DMS production (e.g. Uhlig et al., 2019; Townsend and Keller, 1996; Anderson et al., 2001; Deppeler and Davidson, 2017).

Line 41-42. Not exactly. Zooplankton graze on phytoplankton, which releases DMSP/DMS. Dacey and Wakeham, 1986

L46 This sentence has been changed to make our point more clear.

The CNRM-ESM2-1 and NorESM2-LM models use a prognostic approach, closely related to zooplankton and dimethylsulfoniopropionate abundance, which is a precursor of oceanic DMS (Seland et al., 2020; Séférian et al., 2019)

Line 46. Also mention Hulswar et al 2022 here

L52 done

Line 54. This seems a good place to introduce the flux equation = K * deltaC

L61 We have added in the equation below:

$$DMS_{flux} = K \times \Delta C = K (DMS_w - DMS_a) \hspace{3cm} (1)$$

$\Delta C$ represents the concentration gradient across the air-sea interface where $DMS_w\alpha$ is the concentration of DMS in water, and $DMS_a$ is the concentration in the air but is negligible as this concentration is substantially smaller than that of seawater.

Line 121. By how much? Be quantitative

L132: We have added to this sentence:

In general, the MODIS-aqua Ocean Color chl-a retrieval underestimates Southern Ocean chlorophyll concentrations by up to 25% (Zeng et al., 2016; Haëntjens et al., 2017; Jena, 2017; Gregg and Casey, 2007; Johnson et al., 2013).

Line 126. Liss and Merlivat is piece-wise linear, to be precise

L137: his has been added to the sentence.

LM86 is a piece-wise linear equation and the default parameterization within UKESM1 (Sellar et al., 2019) and was evaluated in combination with all oceanic DMS data sets.

Line 139. Indicate unit for T

L146 T has already been defined previously on line 136 as:

The Schmidt number represents the viscosity/diffusion properties of a gas, varying with respect to sea surface temperature (T in °C).

Eq.4 this is an unusual way of representing Liss and Merlivat 1986, and I'm not sure that they're correct.

Equation for U <3.6 m/s is ok.

For 3.6<U10<13 m/s, it should be (2.85U10-9.65)*(600/Sc)^1/2

For U10>13 m/s, it should be (5.8U10-49.3)* 600/Sc)^1/2

L150 We have now changed the formula in the manuscript to as written above.

Section 2.4.1. please be specific when talking about datasets. Are they datasets of seawater DMS concentration, atmospheric concentration, or both (+fluxes)?

L164 They are just seawater DMS and atmospheric DMS concentrations that we use. We have changed the section title to read:

2.4.1 Oceanic DMS and Atmospheric DMS Datasets

Line 178. 'daily-averaged observations' of CDNC from MODIS?

Daily-averaged CDNC observations derived from MODIS data (Grosvenor et al. 2018; Bennartz & Rausch 2018) were used to calculate CDNC values, based on MODIS satellite data.

Line 180. Here and elsewhere. Phases such as 'we used Choudhury and Tesche (2023)' are too colloquial and inexact. Do the authors mean. E.g. 'we used observations from Choudhury and Tesche (2023)'?

L190 Thank you for the suggestion, this has been changed now to read it as:

Finally, to evaluate cloud condensation nuclei (CCN), we used observations from Choudhury and Tesche (2023) at 818 m, in comparison with simulated CCN at 800 m.

Line 202, 223, 249 etc. Fig number not specified

These have now been fixed

Line 248. 'aligns well' seems generous. MODIS-DMS clearly underestimates relative to observations for the TAN cruise. For SOAP, MODIS-DMS does cover a large range of variability, but its mean is also overestimated.

L258 Based on this comment, we have altered the wording of the sentence:

Having established that oceanic DMS from the MODIS-DMS simulation compares reasonably with summertime observational voyages as seen in Figure 3, 4, we now assess the sensitivity of atmospheric DMS to various sea-to-air transfer functions (Figure 5, A4).

Line 250. How's 'Southern Ocean' defined here?

L31 The definition of the Southern Ocean has been added to the beginning of the introduction.

The representation of aerosols over the Southern Ocean (40 °S to 60 °S) is a large source of uncertainty in climate models due to the lack of observational data and large seasonal variability (Revell et al., 2019).

Line 253, 266, 268. By 'source', it's more exact to say seawater concentration

Where 'source' has been used, has been changed to 'oceanic DMS data set'.

Line 271. Why are the authors choosing to compare their flux estimates with Webb et al. 2019 observations, which were from the coastal zone? The comparison is bound to be poor.

Line 270-283. I don't find these comparisons very insightful, because 1) these previous observation-based flux estimates probably used different K parameterizations (would be slightly more useful to just compare seawater DMS concentration), and 2) many of these observations seem outside the 10-

year simulation window, so I don't know how the authors are able to make direct comparisons (e.g. how were SST, U10, etc treated?).

We have deleted those paragraphs comparing our simulations with previous observations between lines 270 to 283.

Line 286. References here should really be consistent with line 60.

We have added Yang et al., 2011 so that these references are consistent with line 60.

Line 298. Not clear to me what insights these comparisons offer exactly. That variability is greater near the coast? Or high DMS concentrations are associated with sea ice? Keep in mind due to atmospheric transport and loss, atmospheric DMS and seawater DMS concentrations usually correlate very poorly. Here atmospheric DMS measured/modelled at the coast mostly originated from further upwind.

We believe keeping the comparisons between observations across the literature and our simulations from our manuscript would provide insight into how variable the high latitude Southern Ocean atmospheric DMS can be during the summer. As we compare our simulations with observational stations at these higher latitudes provides useful multi-year variability comparisons to our simulations. We show that most oceanic DMS data sets produce much higher atmospheric DMS concentrations over these latitudes than what has been observed. Additionally, we show that using a transfer velocity derived from DMS observations also has better comparisons to the observations than using the other transfer velocities. It also allows an insight into how the model performs during the summer, when sea ice melt occurs, relative to what has been observed.

Line 311, 318. Fig number?

Fixed

Line 320, 321. If I understand correctly, the authors compared MODIS-DMS vs. a climatology of MODIS-DMS (e.g. ~climatology of Chla). To drive flux, in both sets of simulations the other parameters (wind speed, SST) were allowed to be time-varying and contained interannual variability. If so, this comparison underestimates the total interannual variability in DMS flux. The high r2 (0.92) is surely driven in part by the fact that both sets of simulations used the same wind speed, SST, etc. To estimate the total interannual variability in DMS flux, one should probably compare against a climatology of FLUX driven by DMS-MODIS.

The extra simulation shown in Figure 7 is in response to reviewer 1, which was to evaluate the difference induced by the choice of surface seawater DMS data set. We aimed to retain all parameters (SST, wind, etc) to assess the differences of using a climatology of oceanic DMS vs time-series from this parameterisation. This analysis evaluates different years of biological activity (oceanic DMS blooms) and whether there are statistical differences to DMS in the atmosphere compared with a climatology.

Line 329 the wind?

L351 Changed to:

The oceanic DMS signal in the atmosphere is strong but includes large fluctuations driven by the wind variations.

Line 332. This seems in contrast to results from Revell et al. 2019 (line 30). Not quite sure about the phrase 'little change' here. Can authors show panels a c e in Fig. 9 on linear, rather than log scale? By eye there seems to be significant change in at least Nd and CCN. Just because the model (still) severely underestimates Nd and CCN relative to observations (line 339), it doesn't mean that the DMS emission doesn't matter. But rather (probably) the model is missing some other aerosol sources in the Southern Ocean.

Also, would be useful to see maps of simulated Nd/CCN/AOD vs. satellite derived observations. Given the fact that climatology and MODIS-DMS yield very different maps of seawater DMS concentrations, can comparisons with maps of Nd/CCN/AOD yield some clues about which is more realistic?

L354 This paragraph has been altered to include the spread from our simulation, consistent with the previous sections in the manuscript. We decided to retain the log y scale as it better captures the entire distribution of all data sets, including the higher values, rather than the linear scale.

We have also added a new figure for the supplementary materials (Figure S6) showing the DJF spatial distribution of all the oceanic DMS data sets and the observations for AOD, CCN, and CDNC.

It now reads:

Figure 9 and Figure S6 show the effect on cloud and aerosol properties of changing the atmospheric DMS distribution. Changing the atmospheric DMS concentration yields a spread across all our simulations for AOD, CDNC, and CCN by 6%, 15%, and 11%, respectively, over the austral summer Southern Ocean. As DMS predominately oxidises into sulfate within the smaller aerosol modes, it has a smaller influence on the AOD than the larger modes from sea-salt aerosol (Mulcahy et al., 2020). However, these smaller aerosols influence cloud seeding as our simulations show. Changes to the oceanic DMS data set increase the spread in simulated CCN and CDNC over the Southern Ocean rather than changing the DMS emissions, which is consistent with our findings for atmospheric DMS concentrations. Altering the oceanic DMS data set produces a 73% greater change in AOD than altering the DMS emissions over the Southern Ocean, emphasizing the role of the ocean in producing atmospheric DMS. Box plots of AOD, CCN, and CDNC (Figure 9e, a, c) show that the simulations do not capture the maxima in CDNC, CCN or AOD over the Southern Ocean.

[Figure]

*Figure S6. Spatial distribution of each oceanic DMS data set using the Liss and Merlivat (1986) transfer velocity parameterization for (a – d) AOD, (f – i) CDNC, and (k – n) CCN. The observations are from (e) MODIS AOD satellite retrieval, (j) Grosvenor et al. (2018), and (o) the from Choudhury and Tesche (2023).*

Line 335. What do authors mean by 'changing the mean DMS emissions'?

We meant changing the DMS emissions, with the 'mean' now removed from the manuscript.

Line 336. I don't understand what this sentence means

L362 This sentence has been changed to:

Altering the oceanic DMS data set produces a 73% greater change in AOD than altering the DMS emissions over the Southern Ocean, emphasizing the role of the ocean in producing atmospheric DMS.

Line 340. Again, I think it's more accurate to use 'seawater DMS concentration' instead of 'source' here, as to many 'source' = emission

Done

Line 343-344. This sentence makes it sound like the MODIS-DMS is the truth. Probably better to just say that MODIS-DMS simulates lower DMS concentration

This sentence has been removed as part of re-writing the conclusion section.

Line 349. If this is the case, why not just use the climatology (e.g. Hulswar)?

Although our analysis suggests that using a climatology instead of time-series for oceanic DMS may not provide differences across large scale, we also show that capturing spatial variability of oceanic DMS is very important. Therefore, careful consideration of the spatial variability from the climatology is needed. We demonstrate the usefulness of using chlorophyll-a satellite retrieval within an oceanic DMS algorithm to capture a high spatial variability within the Southern Ocean. Although we do suggest that using the newly developed climatology may present good comparisons with observations.

Line 352. 'good spatial representation' doesn't seem like the best choice of words, as MODIS-DMS (Fig 2 b) looks very different from the Lana and Hulswar climatology. Do the authors feel that MODIS-DMS give a more realistic representation of the spatial distribution of DMS than the climatology? And if so, what's their evidence?

L384 We have changed this sentence to more appropriately highlight MODIS-DMS:

We show how using chlorophyll-a data from the MODIS-aqua satellites offers an alternative spatial representation of oceanic DMS based on the chlorophyll-a distribution.

Line 353-356. This sentence could be moved to the beginning of conclusion, as justification for this work

This sentence has now been moved to the beginning of conclusion.

Line 357, 359, 361. 'transfer velocity parametrization', not 'flux parametrization''

Done

Line 358. 'transfer velocity parametrization', not 'sea-to-air parametrization'

Done

Line 363. 'transfer velocity parametrization', not 'DMS parametrization'

Done

Line 366. Specify what these atmospheric DMS concentrations are, for the entire southern ocean? During which months/years? Or during the comparison periods?

'During austral summer over the Southern Ocean' has been added to the beginning of the paragraph.

Line 367. Again, be specific with language. 'Transfer velocity parametrization' is more appropriate than 'DMS-specific relationships'

L412 We have changed this sentence to read:

In future, we recommend that models use up-to-date transfer velocity parametrization specific to DMS such as Blomquist et al. (2017).

I can't find the appendix.

We apologise that you couldn't find the appendix. Although we attached the supplementary materials with the document upon uploading these revisions.